# Representation Learning Beyond Linear Prediction Functions

**Ziping Xu**
Department of Statistics
University of Michigan
zipingxu@umich.edu

**Ambuj Tewari**
Department of Statistics
University of Michigan
tewaria@umich.edu

## Abstract

Recent papers on the theory of representation learning has shown the importance of a quantity called diversity when generalizing from a set of source tasks to a target task. Most of these papers assume that the function mapping shared representations to predictions is linear, for both source and target tasks. In practice, researchers in deep learning use different numbers of extra layers following the pretrained model based on the difficulty of the new task. This motivates us to ask whether diversity can be achieved when source tasks and the target task use different prediction function spaces beyond linear functions. We show that diversity holds even if the target task uses a neural network with multiple layers, as long as source tasks use linear functions. If source tasks use nonlinear prediction functions, we provide a negative result by showing that depth-1 neural networks with ReLu activation function need exponentially many source tasks to achieve diversity. For a general function class, we find that eluder dimension gives a lower bound on the number of tasks required for diversity. Our theoretical results imply that simpler tasks generalize better. Though our theoretical results are shown for the global minimizer of empirical risks, their qualitative predictions still hold true for gradient-based optimization algorithms as verified by our simulations on deep neural networks.

## 1 Introduction

It has become a common practice (Tan et al., 2018) to use a pre-trained network as the representation for a new task with *small* sample size in various areas including computer vision (Marmanis et al., 2015), speech recognition (Dahl et al., 2011; Jaitly et al., 2012; Howard and Ruder, 2018) and machine translation (Weng et al., 2020). Most representation learning is based on the assumption that the source tasks and the target task share the same low-dimensional representation.

In this paper, following the work of Tripuraneni et al. (2020), we assume that each task, indexed by $t$, generates observations noisily from the mean function $f_t^* \circ h^*$, where $h^*$ is the true representation shared by all the tasks and $f_t^*$ is called the prediction function. We assume $h^* \in \mathcal{H}$, the representation function space and $f_t \in \mathcal{F}_t$, the prediction function space. Tripuraneni et al. (2020) proposed a diversity condition which guarantees that the learned representation will generalize to target tasks with any prediction function. Assume we have $T$ source tasks labeled by $1, \dots, T$ and $\mathcal{F}_1 = \cdots = \mathcal{F}_T = \mathcal{F}_{so}$. We denote the target task by $ta$ and let $\mathcal{E}_t(f, h) \in \mathbb{R}$ be the excess error of $f, h \in \mathcal{F}_t \times \mathcal{H}$ for task $t$. The diversity condition can be stated as follows: there exists some $\nu > 0$, such that for any $f_{ta}^* \in \mathcal{F}_{ta}$ and any $h \in \mathcal{H}$,

$$\underbrace{\inf_{f_{ta} \in \mathcal{F}_{ta}} \mathcal{E}_{ta}(f_{ta}, h)}_{\text{Excess error given } h \text{ for target task}} \leq \nu \underbrace{\inf_{f_t \in \mathcal{F}_{so}, t=1,\dots,T} \frac{1}{T} \sum_{t=1}^{T} \mathcal{E}_t(f_t, h)}_{\text{Excess error given } h \text{ for source tasks}}.$$

35th Conference on Neural Information Processing Systems (NeurIPS 2021).

The diversity condition relates the excess error for source tasks and the target task with respect to any fixed representation $h$. A smaller $\nu$ indicates a better transfer from source tasks to the target task. Generally, we say $T$ source tasks achieve diversity over $\mathcal{F}_{ta}$ when $\nu$ is finite and relatively small.

The number of source tasks plays an important role here. To see this, assume $\mathcal{F}_{so} = \mathcal{F}_{ta} = \mathcal{F}$ is a discrete function space and each function can be arbitrarily different. In this case, we will need the target task to be the same as at least one source task, which in turn requires $T \geq |\mathcal{F}|$ and $\nu \geq |\mathcal{F}|$. So far, it has only been understood that when *both* source tasks and target task use *linear* prediction functions, it takes at least $d$ source tasks to be diverse, where $d$ is the number of dimension of the linear mappings. This paper answers the following two open questions with a focus on the deep neural network (DNN) models:

*How does representation learning work when $\mathcal{F}_{so} \neq \mathcal{F}_{ta}$?*
*How many source tasks do we need to achieve diversity with nonlinear prediction functions?*

There are strong practical motivations to answer the above two questions. In practice, researchers use representation learning despite the difference in difficulty levels of source and target tasks. We use a more complex function class for substantially harder target task, which means $\mathcal{F}_{so} \neq \mathcal{F}_{ta}$. This can be reflected as extra layers when a deep neural network model is used as prediction functions. On the other hand, the source task and target task may have different objectives. Representation pretrained on a classification problem, say ImageNet, may be applied to object detection or instance segmentation problems. For instance, Oquab et al. (2014) trained a DNN on ImageNet and kept all the layers as the representation except for the last linear mapping, while two fully connected layers are used for the target task on object detection.

Another motivation for our work is the mismatch between recently developed theories in representation learning and the common practice in empirical studies. Recent papers on the theory of representation learning all require multiple sources tasks to achieve diversity so as to generalize to any target task in $\mathcal{F}$ (Maurer et al., 2016; Du et al., 2020; Tripuraneni et al., 2020). However, most pretrained networks are only trained on a single task, for example, the ImageNet pretrained network. To this end, we will show that a single multi-class classification problem can be diverse.

Lastly, while it is common to simply use linear mapping as the source prediction function, there is no clear theoretical analysis showing whether or not diversity can be achieved with *nonlinear* prediction function spaces.

**Main contributions.** We summarize the main contributions made by this paper.

1. We show that diversity over $\mathcal{F}_{so}$ implies diversity over $\mathcal{F}_{ta}$, when both $\mathcal{F}_{so}$ and $\mathcal{F}_{ta}$ are DNNs and $\mathcal{F}_{ta}$ has more layers. More generally, the same statement holds when $\mathcal{F}_{ta}$ is more complicated than $\mathcal{F}_{so}$, in the way that $\mathcal{F}_{ta} = \mathcal{F}'_{ta} \circ (\mathcal{F}_{so}^{\otimes m})$ for some positive integer $m$[1] and function class $\mathcal{F}'_{ta}$.

2. Turning our attention to the analysis of diversity for non-linear prediction function spaces, we show that for a depth-1 NN, it requires $\Omega(2^d)$ many source tasks to establish diversity with $d$ being the representation dimension. For general $\mathcal{F}_{so}$, we provide a lower bound on the number of source tasks required to achieve diversity using the eluder dimension (Russo and Van Roy, 2013) and provide a upper bound using the generalized rank (Li et al., 2021).

3. We show that, from the perspective of achieving diversity, a single source task with multiple outputs can be equivalent to multiple source tasks. While our theories are built on empirical risk minimization, our simulations on DNNs for a multi-variate regression problem show that the qualitative predictions our theory makes still hold when stochastic gradient descent is used for optimization.

## 2    Preliminaries

We first introduce the mathematical setup of the problem studied in this paper along with the two-phase learning method that we will focus on.

---

[1] $\mathcal{F}^{\otimes T}$ is the $T$ times Cartesian product of $\mathcal{F}$.

**Problem setup.** Let $\mathcal{X}$ denote the input space. We assume the same input distribution $P_X$ for all tasks, as covariate shift is not the focus of this work. In our representation learning setting, there exists a generic feature representation function $h^* \in \mathcal{H} : \mathcal{X} \mapsto \mathcal{Z}$ that is shared across different tasks, where $\mathcal{Z}$ is the feature space and $\mathcal{H}$ is the representation function space. Since we only consider the different prediction functions, each task, indexed by $t$, is defined by its prediction function $f_t^* \in \mathcal{F}_t : \mathcal{Z} \mapsto \mathcal{Y}_t \subset [0,1]$, where $\mathcal{F}_t$ is the prediction function space of task $t$ and $\mathcal{Y}_t$ is the corresponding output space. The observations $Y_t = f_t^* \circ h^*(X) + \epsilon$ are generated noisily with mean function $f_t^* \circ h^*$, where $X \sim P_X$ and $\epsilon$ is zero-mean noise that is independent of $X$.

Our representation is learned on source tasks $\boldsymbol{f}_{so}^* := (f_1^*, \ldots, f_T^*) \in \mathcal{F}_1 \times \cdots \times \mathcal{F}_T$ for some positive integer $T$. We assume that all the prediction function spaces $\mathcal{F}_t = \mathcal{F}_{so}$ are the same over the source tasks. We denote the target task by $f_{ta}^* \in \mathcal{F}_{ta} : \mathcal{Z} \mapsto \mathcal{Y}_t \subset [0,1]$, where $\mathcal{F}_{ta}$ is the target prediction function space. Unlike the previous papers (Du et al., 2020; Tripuraneni et al., 2020; Maurer et al., 2016), which all assume that the same prediction function space is used for all tasks, we generally allow for the possibility that $\mathcal{F}_{so} \neq \mathcal{F}_{ta}$.

**Learning algorithm.** We consider the same two-phase learning method as in Tripuraneni et al. (2020). In the first phase (the training phase), $\boldsymbol{n} = (n_1, \ldots, n_T)$ samples from each task are available to learn a good representation. In the second phase (the test phase), we are presented $n_{ta}$ samples from the target task to learn its prediction function using the pretrained representation learned in the training phase.

We denote a dataset of size $n$ from task $f_t$ by $S_t^n = \{(x_{ti}, y_{ti})\}_{i=1}^n$. We use empirical risk minimization (ERM) for both phase. In the training phase, we minimize average risks over $\{S_t^{n_t}\}_{t=1}^T$:

$$\hat{R}(\boldsymbol{f}, h \mid \boldsymbol{f}_{so}^*) := \frac{1}{\sum_t n_t} \sum_{t=1}^T \sum_{i=1}^{n_t} l_{so}(f_t \circ h(x_{ti}), y_{ti}),$$

where $l_{so} : \mathcal{Y} \times \mathcal{Y} \mapsto \mathbb{R}$ is the loss function for the source tasks and $\boldsymbol{f} = (f_1, \ldots, f_T) \in \mathcal{F}_{so}^{\otimes T}$. The estimates are given by $(\hat{\boldsymbol{f}}_{so}, \hat{h}) \in \arg\min_{\boldsymbol{f}, h} \hat{R}(\boldsymbol{f}, h \mid \boldsymbol{f}_{so}^*)$. In the second phase, we obtain the dataset $\{x_{ta,i}, y_{ta,i}\}_i^{n_{ta}}$ from the target task and our predictor $\hat{f}_{ta}$ is given by

$$\arg\min_{f \in \mathcal{F}_{ta}} \hat{R}(f, \hat{h} \mid f_{ta}^*) := \frac{1}{n_{ta}} \sum_{i=1}^{n_{ta}} l_{ta}(f \circ \hat{h}(x_{ta,i}), y_{ta,i}),$$

for some loss function $l_{ta}$ on the target task. We also use $R(\cdot, \cdot \mid \cdot)$ for the expectation of the above empirical risks. We denote the generalization error of certain estimates $f, h$ by $\mathcal{E}(f, h \mid \cdot) := R(f, h \mid \cdot) - \min_{f' \in \mathcal{F}, h' \in \mathcal{H}} R(f', h' \mid \cdot)$, where $\mathcal{F}$ can be either $\mathcal{F}_{so}^{\otimes T}$ or $\mathcal{F}_{ta}$ depending on the tasks. Our goal is to bound the generalization error of the target task.

For simplicity, our results are presented under square loss functions. However, we show that our results can generalize to different loss functions in Appendix G. We also assume $n_1 = \cdots = n_T = n_{so}$ for some positive integer $n_{so}$.

## 2.1 Model complexity

As this paper considers general function classes, our results will be presented in terms of the complexity measures of classes of functions. We follow the previous literature (Maurer et al., 2016; Tripuraneni et al., 2020) which uses Gaussian complexity. Note that we do not use the more common Rademacher complexity as the proofs require a decomposition theorem that only holds for Gaussian complexity.

For a generic vector-valued function class $\mathcal{Q}$ containing functions $q : \mathbb{R}^d \mapsto \mathbb{R}^r$ and $N$ data points, $X_N = (\boldsymbol{x}_1, \ldots, \boldsymbol{x}_N)^T$, the empirical Gaussian complexity is defined[2] as

$$\hat{\mathfrak{G}}_N(\mathcal{Q}) = \mathbb{E}_{\mathbf{g}} \left[ \sup_{\mathbf{q} \in \mathcal{Q}} \frac{1}{\sqrt{N}} \sum_{i=1}^N g_i^T q(\boldsymbol{x}_i) \right], \quad g_i \sim \mathcal{N}(0, I_r) \quad i.i.d.$$

---

[2]Note that the standard definition has a $1/N$ factor instead of $1/\sqrt{N}$. We use the variant so that $\hat{\mathfrak{G}}_N$ does not scale with $N$ in most of the cases we consider and only reflects the complexity of the class.

The corresponding population Gaussian complexity is defined as $\mathfrak{G}_N(\mathcal{Q}) = \mathbb{E}_{X_N}\left[\hat{\mathfrak{G}}_N(\mathcal{Q})\right]$, where the expectation is taken over the distribution of $X_N$.

## 2.2 Diversity

Source tasks have to be diverse enough, to guarantee that the representation learned from source tasks can be generalized to any target task in $\mathcal{F}_{ta}$. To measure when transfer can happen, we introduce the following two definitions, namely that of *transferability* and *diversity*.

**Definition 1.** *For some $\nu, \mu > 0$, we say the source tasks $f_1^*, \ldots, f_T^* \in \mathcal{F}_{so}$ are $(\nu, \mu)$-transferable to task $f_{ta}^*$ if*

$$\sup_{h \in \mathcal{H}} \frac{\inf_{f \in \mathcal{F}_{ta}} \mathcal{E}(f, h \mid f_{ta}^*)}{\inf_{\boldsymbol{f} \in \mathcal{F}_{so}^{\otimes T}} \mathcal{E}(\boldsymbol{f}, h \mid \boldsymbol{f}_{so}^*) + \mu/\nu} \leq \nu.$$

*Furthermore, we say they are $(\nu, \mu)$-diverse over $\mathcal{F}_{ta}$ if above ratio is bounded for any true target prediction functions $f_{ta}^* \in \mathcal{F}_{ta}$, i.e.*

$$\sup_{f_{ta}^* \in \mathcal{F}_{ta}} \sup_{h \in \mathcal{H}} \frac{\inf_{f \in \mathcal{F}_{ta}} \mathcal{E}(f, h \mid f_{ta}^*)}{\inf_{\boldsymbol{f} \in \mathcal{F}_{so}^{\otimes T}} \mathcal{E}(\boldsymbol{f}, h \mid \boldsymbol{f}_{so}^*) + \mu/\nu} \leq \nu.$$

When it is clear from the context, we denote $\mathcal{E}(f, h \mid f_{ta}^*)$ and $\mathcal{E}(\boldsymbol{f}, h \mid \boldsymbol{f}_{so}^*)$ by $\mathcal{E}_{ta}(f, h)$ and $\mathcal{E}_{so}(\boldsymbol{f}, h)$, respectively. We will call $\nu$ the transfer component and $\mu$, the bias introduced by transfer. The definition of transferable links the generalization error between source tasks and the target task as shown in Theorem 1. The proof can be found in Appendix A.

**Theorem 1.** *If source tasks $f_1^*, \ldots, f_T^*$ are $(\nu, \mu)$-diverse over $\mathcal{F}_{ta}$, then for any $f_{ta}^* \in \mathcal{F}_{ta}$, we have*

$$\mathcal{E}_{ta}(\hat{f}_{ta}, \hat{h}) \leq \nu \mathcal{E}_{so}(\hat{\boldsymbol{f}}_{so}, \hat{h}) + \mu + \frac{\sqrt{2\pi}\hat{\mathfrak{G}}_{n_{ta}}\left(\mathcal{F}_{ta} \circ \hat{h}\right)}{\sqrt{n_{ta}}} + \sqrt{\frac{9\ln(2/\delta)}{2n_{ta}}}.$$

The first term in Theorem 1 can be upper bounded using the standard excess error bound of Gaussian complexity. The benefit of representation learning is due to the decrease in the third term from $\hat{\mathfrak{G}}_{n_{ta}}(\mathcal{F}_{ta} \circ \mathcal{H})/\sqrt{n_{ta}}$ without representation learning to $\hat{\mathfrak{G}}_{n_{ta}}(\mathcal{F}_{ta} \circ \hat{h})/\sqrt{n_{ta}}$ in our case. For the problem with complicated representations, the former term can be extremely larger than the later one.

In the rest of the paper, we discuss when we can bound $(\nu, \mu)$, for nonlinear and nonidentical $\mathcal{F}_{so}$ and $\mathcal{F}_{ta}$.

## 3 Negative transfer when source tasks are more complex

Before introducing the cases that allow transfer, we first look at a case where transfer is impossible.

Let the source task use a linear mapping following the shared representation and the target task directly learns the representation. In other words, $f_{ta}^*$ is identical mapping and known to the learner. We further consider $\mathcal{F}_{so} = \{z \mapsto w^T z : w \in \mathbb{R}^p\}$ and $\mathcal{H} = \{x \mapsto Hx : H \in \mathbb{R}^{p \times d}\}$. The interesting case is when $p \ll d$. Let the optimal representation be $H^*$ and the true prediction function for each source task be $w_1^*, \ldots, w_T^*$. In the best scenario, we assume that there is no noise in the source tasks and each source task collects as many samples as possible, such that we will have an accurate estimation on each $w_t^{*T} H^* \in \mathbb{R}^{p \times d}$ which we denote by $W_t^*$.

However, the hypothesis class given the information from source tasks are

$$\{H \in \mathbb{R}^{p \times d} : \exists w \in \mathbb{R}^p, w^T H = W_t \text{ for all } t = 1, \ldots, T\}.$$

As $H^*$ is in the above class, any $QH^*$ for some rotation matrix $Q \in \mathbb{R}^{p \times p}$ is also in the class. In other words, a non-reducible error of learnt representation is $\max_Q \|H^* - QH^*\|_2^2$ in the worst case.

More generally, we give a condition for negative transfer. Consider a single source task $f_{so}^*$. Let $\mathcal{H}_{so}^* := \arg\min_{h \in \mathcal{H}} \min_{f \in \mathcal{F}_{so}} \mathcal{E}_{so}(f, h)$ and $\mathcal{H}_{ta}^* := \arg\min_{h \in \mathcal{H}} \min_{f \in \mathcal{F}_{ta}} \mathcal{E}_{ta}(f, h)$. In fact, if one hopes to get a representation learning benefit with zero bias ($\mu = 0$), we need all $h \in \mathcal{H}_{so}^*$ to satisfy $\inf_{f \in \mathcal{F}_{ta}} \mathcal{E}_{ta}(f, h) = 0$, i.e. any representation that is optimal in the source task is optimal in the target task as well. Equivalently, we will need $\mathcal{H}_{so}^* \subset \mathcal{H}_{ta}^*$. As shown in Proposition 1, the definition of *transferable* captures the case well.

**Proposition 1.** *If there exists a $h' \in \mathcal{H}_{so}^*$ such that $h' \notin \mathcal{H}_{ta}^*$, then $\min_{f_{ta}} \mathcal{E}_{ta}(f_{ta}, h') > 0$. Furthermore, there is no $\nu < \infty$, such that $f_{so}^*$ is $(\nu, 0)$-transferable to $f_{ta}^*$.*

*Proof.* The first statement is by definition. Plugging $g'$ into the Definition 1, we will have $\nu = \infty$ □

The negative transfer happens when the optimal representation for source tasks may not be optimal for the target task. As the case in our linear example, this is a result of more complex $\mathcal{F}_{so}$, which allows more flexibility to reduce errors. This inspires us to consider the opposite case where $\mathcal{F}_{ta}$ is more complex than $\mathcal{F}_{so}$.

## 4   Source task as a representation

Before discussing more general settings, we first consider a single source task, which we refer to as *so*. Assume that the source task itself is a representation of the target task. Equivalently, the source task has a known prediction function $f_{so}^*(x) = x$. This is a commonly-used framework when we decompose a complex task into several simple tasks and use the output of simple tasks as the input of a higher-level task.

A widely-used transfer learning method, called offset learning, which assumes $f_{ta}^*(x) = f_{so}^*(x) + w_{ta}^*(x)$ for some offset function $w_{ta}^*$ falls within the framework considered here. The offset method enjoys its benefits when $w_{ta}$ has low complexity and can be learnt with few samples. It is worth mentioning that our setting covers a more general setting in Du et al. (2017), which assumes $f_{ta}^*(x) = G(f_{so}^*(x), w_{ta}^*(x))$ for some known transformation function $G$ and unknown $w_{ta}^*$.

We show that a simple Lipschitz condition on $\mathcal{F}_{ta}$ gives us a bounded transfer-component.

**Assumption 1** (Lipschitz assumption). *Any $f_{ta} \in \mathcal{F}_{ta}$ is L-Lipschitz with respect to $L_2$ distance.*

**Theorem 2.** *If Assumption 1 holds, task $f_{so}^*$ is $(L, 0)$-transferable to task $f_{ta}^*$ and we have with a high probability,*

$$\mathcal{E}_{ta}(\hat{f}_{ta}, \hat{h}) = \tilde{\mathcal{O}}\left( \frac{L\hat{\mathfrak{G}}_{n_{so}}(\mathcal{H})}{\sqrt{n_{so}}} + \frac{\hat{\mathfrak{G}}_{n_{ta}}(\mathcal{F}_{ta} \circ \hat{h})}{\sqrt{n_{ta}}} \right).$$

Theorem 2 bounds the generalization error of two terms. The first term that scales with $1/\sqrt{n_{so}}$ only depends on the complexity of $\mathcal{H}$. Though the second term scales with $1/\sqrt{n_{ta}}$, it is easy to see that $\hat{\mathfrak{G}}_{n_{ta}}(\mathcal{F}_{ta} \circ \hat{h}) = \hat{\mathfrak{G}}_{n_{ta}}(\mathcal{F}_{ta}) \ll \hat{\mathfrak{G}}_{n_{ta}}(\mathcal{F}_{ta} \circ \mathcal{H})$ with the dataset $\{\hat{h}(x_{ta,i})\}_{i=1}^{n_{ta}}$.

### 4.1   General case

Previously, we assume that a single source task is a representation of the target task. Now we consider a more general case: there exist functions in the source prediction space that can be used as representations of the target task. Formally, we consider $\mathcal{F}_{ta} = \mathcal{F}_{ta}' \circ (\mathcal{F}_{so}^{\otimes m})$ for some $m > 0$ and some target-specific function space $\mathcal{F}_{ta}' : \mathcal{Y}_{so}^{\otimes m} \mapsto \mathcal{Y}_{ta}$. Note that $\mathcal{F}_{ta}$ is strictly larger than $\mathcal{F}_{so}$ when $m = 1$ and the identical mapping $x \mapsto x \in \mathcal{F}_{ta}'$. In practice, ResNet (Tai et al., 2017) satisfies the above property.

**Assumption 2.** *Assume any $f_{ta}' \in \mathcal{F}_{ta}'$ is $L'$-Lipschitz with respect to $L_2$ distance.*

**Theorem 3.** *If Assumption 2 holds and the source tasks are $(\nu, \mu)$-diverse over its own space $\mathcal{F}_{so}$, then we have*

$$\mathcal{E}_{ta}\left(\hat{f}_{ta}, \hat{h}\right) = \tilde{\mathcal{O}}\left( L'm\left( \frac{\nu\hat{\mathfrak{G}}_{Tn_{so}}(\mathcal{F}_{so}^{\otimes T} \circ \mathcal{H})}{\sqrt{Tn_{so}}} + \mu \right) + \frac{\hat{\mathfrak{G}}_{n_{ta}}\left(\mathcal{F}_{ta} \circ \hat{h}\right)}{\sqrt{n_{ta}}} \right).$$

It is shown in Tripuraneni et al. (2020) that $\hat{\mathfrak{G}}_{\sum_s n_s}(\mathcal{F}_{so}^{\otimes S} \circ \mathcal{H})$ can be bounded by $\tilde{\mathcal{O}}(\hat{\mathfrak{G}}_{Tn_{so}}(\mathcal{H}) + \sqrt{T}\hat{\mathfrak{G}}_{n_{so}}(\mathcal{F}_{so}))$. Thus, the first term scales with $\hat{\mathfrak{G}}_{Tn_{so}}(\mathcal{H})/\sqrt{Tn_{so}} + \hat{\mathfrak{G}}_{n_{so}}(\mathcal{F}_{so})/\sqrt{n_{so}}$. Again the common part shared by all the tasks decreases with $\sqrt{Tn_{so}}$, while the task-specific part scales with $\sqrt{n_{so}}$ or $\sqrt{n_{ta}}$.

## 4.2 Applications to deep neural networks

Theorem 3 has a broad range of applications. In the rest of the section, we discuss its application to deep neural networks, where the source tasks are transferred to a target task with a deeper network.

We first introduce the setting for deep neural network prediction function. We consider the regression problem with $\mathcal{Y}_{so} = \mathcal{Y}_{ta} = \mathbb{R}$ and the representation space $\mathcal{Z} \subset \mathbb{R}^p$. A depth-$K$ vector-valued neural network is denoted by
$$f(\mathbf{x}) = \sigma\left(\mathbf{W}_K\left(\sigma\left(\ldots\sigma\left(\mathbf{W}_1\mathbf{x}\right)\right)\right)\right),$$
where each $\mathbf{W}_k$ is a parametric matrix of layer $k$ and $\sigma$ is the activation function. For simplicity, we let $\mathbf{W}_k \in \mathbb{R}^{p \times p}$ for $k = 1, \ldots K$. The class of all depth-$K$ neural network is denoted by $\mathcal{M}_K$. We denote the linear class by $\mathcal{L} = \{x \mapsto \alpha^T x + \beta : \forall \alpha \in \mathbb{R}^p, \|\alpha\|_2 \leq M(\alpha)\}$ for some $M(\alpha) > 0$. We also assume
$$\max\{\|W_k\|_\infty, \|W_k\|_2\|\} \leq M(k).$$
where $\|\cdot\|_\infty$ and $\|\cdot\|_2$ are the infinity norm and spectral norm. We assume any $z \in \mathcal{Z}, \|z\|_\infty \leq D_Z$.

**Deeper network for the target task.** We now consider the source task with prediction function of a depth-$K_{so}$ neural network followed by a linear mapping and target task with depth-$K_{ta}$ neural network. We let $K_{ta} > K_{so}$. Then we have
$$\mathcal{F}_{ta} = \mathcal{L} \circ \mathcal{M}_{K_{ta}-K_{so}} \circ \mathcal{M}_{K_{so}} \text{ and } \mathcal{F}_{so} = \mathcal{L} \circ \mathcal{M}_{K_{so}}.$$

Using the fact that $\mathcal{M}_1 = \sigma(\mathcal{L}^{\otimes p})$, we can write $\mathcal{F}_{ta}$ as $\mathcal{L} \circ \mathcal{M}_{K_{ta}-K_{so}-1} \circ \sigma \circ (\mathcal{F}_{so}^{\otimes p})$. Thus, we can apply Theorem 3 and the standard Gaussian complexity bound for DNN models, which gives us Corollary 1.

**Corollary 1.** *Let $\mathcal{F}_{so}$ be depth-$K_{so}$ neural network and $\mathcal{F}_{ta}$ be depth-$K_{ta}$ neural network. If source tasks are $(\nu, 0)$-diverse over $\mathcal{F}_{so}$, we have*

$$\mathcal{E}_{ta}\left(\hat{f}_{ta}, \hat{h}\right) =$$
$$\tilde{\mathcal{O}}\left(p\nu M(\alpha)\Pi_{k=1}^{K_{ta}}M(k)\left(\frac{\hat{\mathfrak{S}}_{Tn_{so}}(\mathcal{H})}{\sqrt{Tn_{so}}} + \frac{D_Z\sqrt{K_{so}}}{\sqrt{n_{so}}}\right) + \frac{D_Z\sqrt{K_{ta}} \cdot M(\alpha)\Pi_{k=1}^{K_{ta}}M(k)}{\sqrt{n_{ta}}}\right). \quad (1)$$

Note that the terms that scales with $1/\sqrt{n_{so}}$ and $1/\sqrt{n_{ta}}$ have similar coefficients $M(\alpha)\Pi_{k=1}^{K_{ta}}M(k)$, which do not depends on the complexity of $\mathcal{H}$. The term that depends on $\hat{\mathfrak{S}}_{Tn_{so}}(\mathcal{H})$ scales with $1/\sqrt{Tn_{so}}$ as we expected.

## 5 Diversity of non-linear function classes

While Corollary 1 considers the nonlinear DNN prediction function space under a diversity condition, it is not clearly understood how diversity can be achieved for nonlinear spaces. In this section, we first discuss a specific non-linear prediction function space and show a fundamental barrier to achieving diversity. Then we extend our result to general function classes by connecting eluder dimension and diversity. We end the section with positive results for achieving diversity under a generalized rank condition.

We consider a subset of depth-1 neural networks with ReLu activation function: $\mathcal{F} = \{x \mapsto [\langle x, w\rangle - (1 - \epsilon/2)]_+ : \|x\|_2 \leq 1, \|w\| \leq 1\}$ for some $\epsilon > 0$. Our lower bound construction is inspired by the similar construction in Theorem 5.1 of Dong et al. (2021).

**Theorem 4.** *Let $\mathcal{T} = \{f_1, \ldots, f_T\}$ be any set of depth-1 neural networks with ReLu activation in $\mathcal{F}$. For any $\epsilon > 0$, if $T \leq 2^{d\log(1/\epsilon)-1}$, there exists some representation $h^*, h' \in \mathcal{H}$, some distribution $P_X$ and a target function $f_{ta}^* \in \mathcal{F}$, such that*
$$\inf_{\boldsymbol{f} \in \mathcal{F}^{\otimes T}} \mathcal{E}_{so}(\boldsymbol{f}, h') = 0, \quad while \quad \inf_{f \in \mathcal{F}} \mathcal{E}_{ta}(f, h') = \epsilon^2/32.$$

Theorem 4 implies that we need at least $\Omega(2^d\log(1/\epsilon))$ source tasks to achieve diversity. Otherwise, we can always find a set of source tasks and a target task such that the generalization error in source tasks are minimized to 0 while that in target task is $\epsilon^2/32$. Though ReLu gives us a more intuitive result, we do show that similar lower bounds can be shown for other popular activation functions, for example, sigmoid function (see Appendix E).

## 5.1 Lower bound using eluder dimension

We extend our result by considering a general function space $\mathcal{F}$ and build an interesting connection between diversity and eluder dimension (Russo and Van Roy, 2013). We believe we are the first to notice a connection between eluder dimension and transfer learning.

Eluder dimension has been used to measure the sample complexity in the Reinforcement Learning problem. It considers the minimum number of *inputs*, such that any two functions evaluated similarly at these inputs will also be similar at any other input. However, diversity considers the minimum number of functions such that any two representations with similar *outputs* for these functions will also have similar output for any other functions. Thus, there is a kind of duality between eluder dimension and diversity.

We first formally define eluder dimension. Let $\mathcal{F}$ be a function space with support $\mathcal{X}$ and let $\epsilon > 0$.

**Definition 2** $((\mathcal{F}, \epsilon)$-dependence and eluder dimension (Osband and Roy (2014))). *We say that $x \in \mathcal{X}$ is $(\mathcal{F}, \epsilon)$-dependent on $\{x_1, \ldots, x_n\} \subset \mathcal{X}$ iff*

$$\forall f, \tilde{f} \in \mathcal{F}, \quad \sum_{i=1}^{n} \left\| f(x_i) - \tilde{f}(x_i) \right\|_2^2 \leq \epsilon^2 \implies \| f(x) - \tilde{f}(x) \|_2 \leq \epsilon.$$

*We say $x \in \mathcal{X}$ is $(\mathcal{F}, \epsilon)$-independent of $\{x_1, \ldots, x_n\}$ iff it does not satisfy the definition of dependence.*

*The eluder dimension $\dim_E(\mathcal{F}, \epsilon)$ is the length of the longest possible sequence of elements in $\mathcal{X}$ such that for some $\epsilon' \geq \epsilon$ every element is $(\mathcal{F}, \epsilon')$-independent of its predecessors.*

*We slightly change the definition and let $\dim_E^s(\mathcal{F}, \epsilon)$ be the shortest sequence such that any $x \in \mathcal{X}$ is $(\mathcal{F}, \epsilon)$-dependent on the sequence.*

Although $dim_E^s(\mathcal{F}, \epsilon) \leq dim_E(\mathcal{F}, \epsilon)$, it can be shown that in many regular cases, say linear case and generalized linear case, $dim_E(\mathcal{F}, \epsilon)$ is only larger then $dim_E^s(\mathcal{F}, \epsilon)$ up to a constant.

**Definition 3** (Dual class). *For any $\mathcal{F} : \mathcal{X} \mapsto \mathcal{Y}$, we call $\mathcal{F}^* : \mathcal{F} \mapsto \mathcal{Y}$ its dual class iff. $\mathcal{F}^* = \{g_x : g_x(f) = f(x), \forall x \in \mathcal{X}\}$.*

**Theorem 5.** *For any function class $\mathcal{F} : \mathcal{X} \mapsto \mathbb{R}$, and some $\epsilon > 0$, let $\mathcal{F}^*$ be the dual class of $\mathcal{F}$. Let $d_E = \dim_E^s(\mathcal{F}^*, \epsilon)$. Then for any sequence of tasks $f_1, \ldots, f_t$, $t \leq d_E - 1$, there exists a task $f_{t+1} \in \mathcal{F}$ such that for some data distribution $P_X$ and two representations $h$, $h^*$,*

$$\frac{\inf_{f'_{t+1} \in \mathcal{F}} \mathbb{E}_X \| f'_{t+1}(h(X)) - f_{t+1}(h^*(X)) \|_2^2}{\frac{1}{t} \inf_{f'_1, \ldots, f'_t} \sum_{i=1}^{t} \mathbb{E}_X \| f'_i(h(X)) - f_i(h^*(X)) \|_2^2} \geq t/2.$$

Theorem 5 formally describes the connections between eluder dimension and diversity. To interpret the theorem, we first discuss what are good source tasks. Any $T$ source tasks that are diverse could transfer well to a target task if the parameter $\nu$ could be bounded by some fixed value that is not increasing with $T$. For instance, in the linear case, $\nu = \mathcal{O}(d)$ no matter how large $T$ is. While for a finite function class, in the worst case, $\nu$ will increase with $T$ before $T$ reaches $|\mathcal{F}|$. Theorem 5 states that if the eluder dimension of the dual space is at least $d_E$, then $\nu$ scales with $T$ until $T$ reaches $d_E$, as if the function space $\mathcal{F}$ is discrete with $d_E$ elements.

Note that this result is consistent with what is shown in Theorem 9 as eluder dimension of the class discussed above is lower bounded by $\Omega(2^d)$ as well (Li et al., 2021).

## 5.2 Upper bound using approximate generalized rank

Though we showed that diversity is hard to achieve in some nonlinear function class, we point out that diversity can be easy to achieve if we restrict the target prediction function such that they can be realized by linear combinations of some known basis. Tripuraneni et al. (2020) has shown that any source tasks $f_1, \ldots, f_T \in \mathcal{F}$ are $(1/T, \mu)$-diverse over the space $\{f \in \mathcal{F} : \exists \tilde{f} \in \text{conv}(f_1, \ldots, f_T)$ such that $\sup_z \| f(z) - \tilde{f}(z) \| \leq \mu\}$, where $\text{conv}(f_1, \ldots, f_T)$ is the convex hull of $\{f_1, \ldots, f_T\}$. This can be characterized by a complexity measure, called generalized rank. Generalized rank is the smallest dimension required to embed the input space such that all hypotheses in a function class can be realizable as halfspaces. Generalized rank has close connections to eluder dimension. As shown in Li et al. (2021), eluder dimension can be upper bounded by generalized rank.

**Definition 4** (Approximate generalized rank with identity activation). *Let $\mathcal{B}_d(R) := \{x \in \mathbb{R}^d \mid \|x\|_2 \leq R\}$. The $\mu$-approximate $\mathrm{id\text{-}rk}(\mathcal{F}, R)$ of a function class $\mathcal{F} : \mathcal{X} \mapsto \mathbb{R}$ at scale $R$ is the smallest dimension $d$ for which there exists mappings $\phi : \mathcal{X} \mapsto \mathcal{B}_d(1)$ and $w : \mathcal{F} \mapsto \mathcal{B}_d(R)$ such that*

$$\text{for all } (x, f) \in \mathcal{X} \times \mathcal{F} : |f(x) - \langle w(f), \phi(x)\rangle| \leq \mu.$$

**Proposition 2.** *For any $\mathcal{F}$ with $\mu$-approximate $\mathrm{id\text{-}rk}(\mathcal{F}, R) \leq d_i$ for some $R > 0$, there exists no more than $d_i$ functions, $f_1, \ldots, f_{d_i}$, such that $w(f_1), \ldots w(f_{d_i})$ span $\mathbb{R}^{d_{d_i}}$. Then $f_1, \ldots, f_{d_i}$ are $(d_i, \mu)$-diverse over $\mathcal{F}$.*

Proposition 2 is a direct application of Lemma 7 in Tripuraneni et al. (2020). Upper bounding id-rank is hard for general function class. However, this notation can be useful for those function spaces with a known set of basis functions. For example, any function space $\mathcal{F}$ that is square-integrable has a Fourier basis. Though the basis is an infinite set, we can choose a truncation level $d$ such that the truncation errors of all functions are less than $\mu$. Then the hypothesis space $\mathcal{F}$ has $\mu$-approximate id-rank less than $d$.

# 6 Experiments

In this section, we use simulated environments to evaluate the actual performance of representation learning on DNNs trained with gradient-based optimization methods. We also test the impact of various hyperparameters. Our results indicate that even though our theory is shown to hold for ERM estimators, their qualitative theoretical predictions still hold when Adam is applied and the global minima might not be found. Before we introduce our experimental setups, we discuss a difference between our theories and experiments: our experiments use a single regression task with multiple outputs as the source task, while our analyses are built on multiple source tasks.

## 6.1 Diversity of problems with multiple outputs

Though we have been discussing achieving diversity from multiple source tasks, we may only have access to a single source task in many real applications, for example, ImageNet, which is also the case in our simulations. In fact, we can show that diversity can be achieved by a single multiclass classification or multivariate regression source task. The diversity for the single multivariate regression problem with L2 loss is trivial as the loss function is decomposable. For multiclass classification problems or regression problems with other loss functions, we will need some assumptions on boundedness and continuous. We refer the readers to Appendix H for details.

## 6.2 Experiments setup

Our four experiments are designed for the following goals. The first two experiments (Figure 1, a and b) target on the actual dependence on $n_{so}, n_{ta}, K_{so}$ and $K_{ta}$ in Equation (1) that upper bounds the errors of DNN prediction functions. Though the importance of diversity has been emphasized by various theories, no one has empirically shown its benefits over random tasks selection, which we explore in our third experiment (Figure 1, c). Our fourth experiment (Figure 1, d) verifies the theoretical negative results of nonlinearity of source prediction functions showed in Theorem 4 and 5.

Though the hyper-parameters vary in different experiments, our main setting can be summarized below. We consider DNN models of different layers for both source and target tasks. The first $K$ layers are the shared representation. The source task is a multi-variate regression problem with output dimension $p$ and $K_{so}$ layers following the representation. The target task is a single-output regression problem with $K_{ta}$ layers following the representation. We used the same number of units for all the layers, which we denote by $n_u$. A representation is first trained on the source task using $n_{so}$ random samples and is fixed for the target task, trained on $n_{ta}$ random samples. In contrast, the baseline method trains the target task directly on the same $n_{ta}$ samples without the pretrained network. We use Adam with default parameters for all the training. We use MSE (Mean Square Error) to evaluate the performance under different settings.

## 6.3 Results

Figure 1 summaries our four experiments, with all the Y axes representing the average MSE's of 100 independent runs with an error bar showing the standard deviation. The X axis varies depending

on the goal of each experiment. In subfigure (a), we test the effects of the numbers of observations for both source and target tasks, while setting other hyperparameters by default values. The X axis represents $n_{so}$ and the colors represents $n_{ta}$. In subfigure (b), we test the effects of the number of shared representation layers $K$. To have comparable MSE's, we keep the sum $K + K_{ta} = 6$ and run $K = 1, \dots, 5$ reflected in the X axis, while keeping $K_{so} = 1$. In subfigure (c), we test the effects of diversity. The larger $p$ we have, the more diverse the source task is. We keep the actual number of observations $n_{so} \cdot p = 4000$ for a fair comparison. Lastly, in subfigure (d), we test whether diversity is hard to achieve when the source prediction function is nonlinear. The X axis is the number of layers in source prediction function $K_{so}$. The nonlinearity increases with $K_{so}$. We run $K_{so} = 1, 2, 3$ and add an activation function right before the output such that the function is nonlinear even if $K_{so} = 1$.

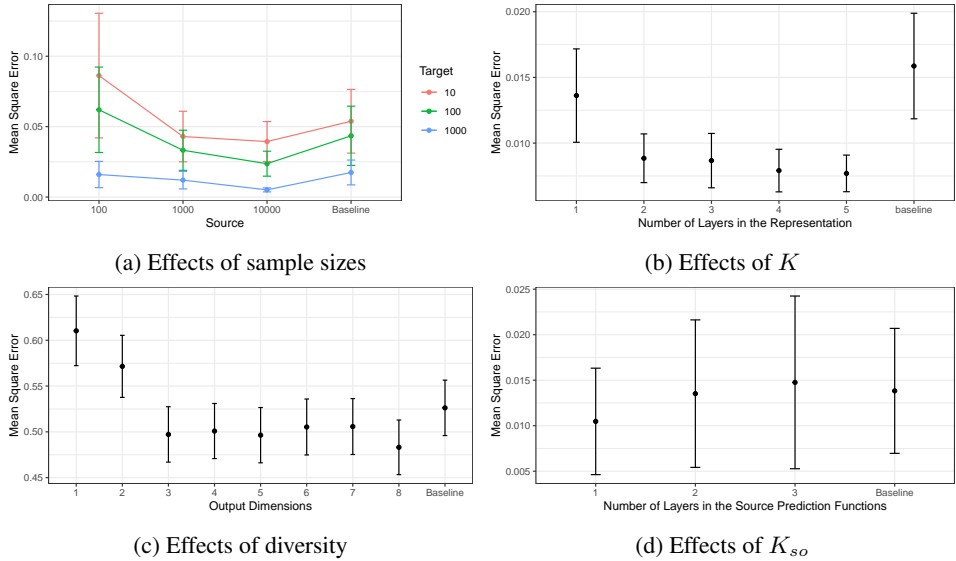

Figure 1: **(a)** Effects of the numbers of observations for both source ($n_{so}$) and target tasks ($n_{ta}$). **(b)** Effects of the number of shared representation layers $K$. **(c)** Effects of diversity determined by the output dimensions $p$. We keep the actual number of observations $n_{so} \cdot p = 4000$. **(d)** Effects of nonlinearity of the source prediction function. Higher $K_{so}$ indicates higher nonlinearity.

In Figure 1 (a), MSE's decrease with larger numbers of observations in both source and target tasks, while there is no significant difference between $n_{so} = 1000$ and $10000$. The baseline method without representation learning performs worst and it performs almost the same when $n_{ta}$ reaches 1000. In (b), there are positive benefits for all different numbers of the shared layers and the MSE is the lowest at 5 shared layers. As shown in Figure 1 (c), the MSE's and their variances are decreasing when the numbers of outputs increase, i.e., higher diversity. Figure 1 (d) shows that there is no significant difference between baseline and $K_{so} = 1, 2, 3$. When $K_{so} = 2, 3$ there is a negative effects.

## 7 Discussion

In this paper, we studied representation learning beyond linear prediction functions. We showed that the learned representation can generalize to tasks with multi-layer neural networks as prediction functions as long as the source tasks use linear prediction functions. We show the hardness of being diverse when the source tasks are using nonlinear prediction functions by giving a lower bound on the number of source tasks in terms of eluder dimension. We further give an upper bound depending on the generalized rank.

Focusing on future work, we need better tools to understand the recently proposed complexity measure generalized rank. Our analyses rely on the ERM, while in practice as well as in our simulations, gradient-based optimization algorithms are used. Further analyses on the benefits of representation learning that align with practice on the choice of optimization method should be studied. Our analyses assume arbitrary source tasks selection, while, in real applications, we may have limited data from

groups that are under-represented, which may lead to potential unfairness. Algorithm that calibrates this potential unfairness should be studied.

Another direction of future work is to propose reasonable assumptions for a guarantee of generalization even when the source prediction functions are nonlinear. Possibly the interesting functions are within a subset of the target prediction function class, which makes generalization easier.

In this work, we consider provided and fixed source tasks. In practice, many empirical methods (Graves et al., 2017) are proposed to adaptively select unknown source tasks. An interesting direction is to ask whether adaptively task selection could achieve the optimal level of diversity and to design algorithms for that purpose.

## 8 Funding Disclosure & Acknowledgements

This work was directly supported by NSF grant IIS-2007055.

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
