## A Proof of Theorem 1

*Proof.* Note that the second phase is to find the best function within the class $\mathcal{F}_{ta} \circ \hat{h}$. We first apply the standard bounded difference inequality (Bartlett and Mendelson, 2002) as shown in Theorem 6.

**Theorem 6** (Bartlett and Mendelson (2002)). *With a probability at least $1 - \delta$,*

$$\sup_{f \in \mathcal{F}_{ta}} |R_{ta}(f \circ \hat{h}) - \hat{R}_{ta}(f \circ \hat{h})| \leq \frac{\sqrt{2\pi}\hat{\mathfrak{G}}_{n_{ta}}(\mathcal{F}_{ta} \circ \hat{h})}{\sqrt{n_{ta}}} + \sqrt{\frac{9\ln(2/\delta)}{2n_{ta}}} =: \epsilon(\hat{h}, n_{ta}, \delta),$$

*furthermore, the total generalization error can be upper bounded by*

$$\mathcal{E}_{ta}(\hat{f}_{ta} \circ \hat{h}) \leq \underbrace{\inf_{f \in \mathcal{F}_{ta}} \mathcal{E}_{ta}(f, \hat{h})}_{\textit{approximation error}} + \underbrace{\epsilon(\hat{h}, n_{ta}, \delta)}_{\textit{generalization error over } \mathcal{F}_{ta} \circ \hat{h}}. \tag{2}$$

Theorem 6 is stated in terms of Gaussian complexity. It is more common to use Radamecher complexity, which can be upper bounded by $\sqrt{2\pi}$ of the corresponding Gaussian complexity. For the generalization bound in terms of Rademacher complexity, Theorem 26.5 of Shalev-Shwartz and Ben-David (2014) has a full proof. Then recall that $\mathcal{Y}_t$ and $\mathcal{Y}_{ta} \subset [0,1]$, we get rid of the loss function by the contraction lemma, which leads to Theorem 6. The result follows by the definition,

$$\inf_{f \in \mathcal{F}_{ta}} \mathcal{E}_{ta}(f, \hat{h}) \leq \frac{\inf_{f_{ta} \in \mathcal{F}_{ta}} \mathcal{E}_{ta}(f_{ta}, \hat{h})}{\inf_{\boldsymbol{f}_{so} \in \mathcal{F}_{so}^{\otimes S}} \mathcal{E}_{so}(\boldsymbol{f}_{so}, \hat{h}) + \mu/\nu} (\mathcal{E}_{so}(\hat{\boldsymbol{f}}_{so}, \hat{h}) + \mu/\nu) \leq \nu \mathcal{E}_{so}(\hat{\boldsymbol{f}}_{so}, \hat{h}) + \mu.$$

$\square$

## B Proof of Theorem 2

*Proof.* To show $f_{so}^*$ is $(L, 0)$-transferable to $f_{ta}^*$, we bound the approximation error of the target task given any fixed $h \in \mathcal{H}$.

$$\begin{aligned}
&\mathcal{E}_{ta}(f_{ta}^*, h) \\
&= \mathbb{E}_{X,Y}\left[l_{ta}(f_{ta}^* \circ h(X), Y) - l_{ta}(f_{ta}^* \circ h^*(X), Y)\right] \\
&= \mathbb{E}_X \|f_{ta}^* \circ h(X) - f_{ta}^* \circ h^*(X)\|_2^2 \\
&\leq L\mathbb{E}_X \|h(X) - h^*(X)\|_2^2.
\end{aligned} \tag{3}$$

Now using Assumption 1, we have

$$\begin{aligned}
\mathcal{E}_{so}(h) &= \mathbb{E}_{X,Y}[l_{so}(h(X), Y) - l_{so}(h^*(X), Y)] \\
&= E_X \|h^*(X) - h(X)\|_2^2
\end{aligned}$$

Combined with Equation (3), we have $\sup_{h \in \mathcal{H}}[\mathcal{E}_{ta}(f_{ta}^*, h)/\mathcal{E}_{so}(h)] \leq L$.

Firstly, using Theorem 6 on source tasks solely, we have $\mathcal{E}_{so}(\hat{h}) = \tilde{\mathcal{O}}(\hat{\mathfrak{G}}_{n_{so}}(\mathcal{G})/\sqrt{n_{so}})$. Definition 1 gives us

$$\inf_{f \in \mathcal{F}_{ta}} \mathcal{E}_{ta}(f, \hat{h}) \leq L\mathcal{E}_{so}(\hat{h}) = \tilde{\mathcal{O}}(L\hat{\mathfrak{G}}_{n_{so}}(\mathcal{G})/\sqrt{n_{so}}).$$

Combined with (2), we have

$$\mathcal{E}_{ta}\left(\hat{f}_{ta}, \hat{h}\right) = \tilde{\mathcal{O}}\left(L\frac{\hat{\mathfrak{G}}_{n_{so}}(\mathcal{G})}{\sqrt{n_{so}}} + \frac{\hat{\mathfrak{G}}_{n_{ta}}\left(\mathcal{F}_{ta} \circ \hat{h}\right)}{\sqrt{n_{ta}}}\right).$$

$\square$

# C Proof of Theorem 3

*Proof.* Let $f_{ta}^*(x) = f_{ta}'^*(f_1^* \circ h^*(x), \dots f_m^* \circ h^*(x))$. Define new tasks $t_1, \dots t_m$. Each $t_i$ has the prediction function $f_i^*$.

By Definition 1, the source tasks are $(\nu, \mu)$-transferable to each $t_i$. By Theorem 1, we have

$$\inf_{f_i \in \mathcal{F}_{so}} \mathcal{E}_{t_i}(f_i, \hat{h}) \leq \nu \mathcal{E}_{so}(\hat{\boldsymbol{f}}_{so}, \hat{h}) + \mu.$$

Since we use $L_2$ loss,

$$\inf_{f_i \in \mathcal{F}_{so}} \mathbb{E}_X \|f_i \circ \hat{h}(X) - f_i^* \circ h^*(X)\|_2^2 = \inf_{f_i \in \mathcal{F}_{so}} \mathcal{E}_{t_i}(f_i, \hat{h}) \leq \nu \mathcal{E}_{so}(\hat{\boldsymbol{f}}_{so}, \hat{h}) + \mu.$$

As this holds for all $i \in [m]$, we have

$$\inf_{f_1, \dots, f_m \in \mathcal{F}_{so}} \mathbb{E}_X \|(f_1 \circ \hat{h}(X), \dots, f_m \circ \hat{h}(X)) - (f_1^* \circ h^*(X), \dots, f_m^* \circ \hat{h}^*(X))\|_2^2 \leq m(\nu \mathcal{E}_{so}(\hat{\boldsymbol{f}}_{so}, \hat{h}) + \mu).$$

Using Assumption 2, we have

$$\inf_{f \in \mathcal{F}_{ta}} \mathcal{E}_{ta}(f, \hat{h}) \leq L'm(\nu \mathcal{E}_{so}(\hat{\boldsymbol{f}}_{so}, \hat{h}) + \mu).$$

Theorem 3 follows by plugging this into (2). $\qquad\square$

# D Proof of Corollary 1

This section we prove Corollary 1 using Theorem 3, the standard bound for Gaussian complexity of DNN model and the Gaussian complexity decomposition from Tripuraneni et al. (2020).

The following theorem bounds the Rademacher complexity of a deep neural network model given an input dataset $\boldsymbol{X}_N = (\boldsymbol{x}_1, \dots, \boldsymbol{x}_N)^T \in \mathbb{R}^{N \times d}$.

**Theorem 7** (Golowich et al. (2018)). *Let $\sigma$ be a 1-Lipschitz activation function with $\sigma(0) = 0$. Recall that $\mathcal{M}_K$ is the depth $K$ neural network with $d$-dimensional output with bounded input $\|x_{ji}\| \leq D_Z$ and $\|W_k\|_\infty \leq M(k)$ for all $k \in [K]$. Recall that $\mathcal{L} = \{x \mapsto \alpha^T x + \beta : \forall \alpha \in \mathbb{R}^p, \|\alpha\|_2 \leq M(\alpha)\}$ is the linear class following the depth-$K$ neural network. Then,*

$$\mathfrak{R}_n(\mathcal{L} \circ M_K; \boldsymbol{X}_N) \leq \frac{2D_Z\sqrt{K + 2 + \log d} \cdot M(\alpha)\Pi_{k=1}^K M(k)}{\sqrt{n}}.$$

Since for any function class $\mathcal{F}$, $\hat{\mathfrak{G}}_n(\mathcal{F}) \leq 2\sqrt{\log n} \cdot \hat{\mathfrak{R}}_n(\mathcal{F})$, we also have the bound for the Gaussian complexity under the same conditions.

Applying Theorem 7, we have an upper bound for the second term in Theorem 3:

$$\frac{\hat{\mathfrak{G}}_{n_{ta}}\left(\mathcal{F}_{ta} \circ \hat{h}\right)}{\sqrt{n_{ta}}} \leq \frac{2D_Z\sqrt{\log(n_{ta})}\sqrt{K_{ta} + 2 + \log(d)}M_\alpha \Pi_{k=1}^{K_{ta}} M(k)}{\sqrt{n_{ta}}} = \tilde{\mathcal{O}}\left(\frac{D_Z sqrt K_{ta} M(\alpha)\Pi_{k=1}^{K_{ta}} M(k)}{\sqrt{n_{ta}}}\right).$$

It only remains to bound $\hat{\mathfrak{G}}_{Tn_{so}}\left(\mathcal{F}_{so}^{\otimes T} \circ \mathcal{H}\right)/\sqrt{Tn_{so}}$ in Theorem 3. To proceed, we introduce the decomposition theorem for Gaussian complexity (Tripuraneni et al., 2020).

**Theorem 8** (Theorem 7 in Tripuraneni et al. (2020)). *Let the function class $\mathcal{F}$ consist of functions that are $L(\mathcal{F})$-Lipschitz and have boundedness parameter $D_X = \sup_{f, f', x, x'} \|f(x) - f'(x')\|_2$. Further, define $\mathcal{Q} = \{h(\bar{X}) : h \in \mathcal{H}, \bar{X} \in \cup_{j=1}^T \{X_j\}\}$. Then the Gaussian complexity of the function class $\mathcal{F}^{\otimes T}(\mathcal{H})$ satisfies,*

$$\hat{\mathfrak{G}}_{\mathbf{X}}\left(\mathcal{F}^{\otimes T}(\mathcal{H})\right) \leq \frac{4D_{\mathbf{X}}}{(nT)^{3/2}} + 128C\left(\mathcal{F}^{\otimes T}(\mathcal{H})\right) \cdot \log(nT),$$

*where $C\left(\mathcal{F}^{\otimes t}(\mathcal{H})\right) = L(\mathcal{F})\hat{\mathfrak{G}}_{\mathbf{X}}(\mathcal{H}) + \max_{\mathbf{q} \in \mathcal{Q}} \hat{\mathfrak{G}}_{\mathbf{q}}(\mathcal{F})$.*

With Theorem 8 applied, we have

$$\frac{\hat{\mathfrak{G}}_{Tn_{so}}\left(\mathcal{F}_{so}^{\otimes T} \circ \mathcal{H}\right)}{\sqrt{Tn_{so}}} \leq \frac{8D_X}{(Tn_{so})^2} + \frac{128\left(L(\mathcal{F}_{so})\hat{\mathfrak{G}}_{Tn_{so}}(\mathcal{H}) + \max_{\mathbf{q}\in\mathcal{Q}}\hat{\mathfrak{G}}_{\mathbf{q}}(\mathcal{F}_{so})\right) \cdot \log(Tn_{so})}{\sqrt{Tn_{so}}}.$$
(4)

The second term relies on the Lipschitz constant of DNN, which we bound with the following lemma. Similar results are given by Scaman and Virmaux (2018); Fazlyab et al. (2019).

**Lemma 1.** *If the activation function is* 1-*Lipschitz, any function in* $\mathcal{L} \circ \mathcal{M}_K$ *is* $M(\alpha)\Pi_{k=1}^K M(k)$-*Lipschitz with respect to* $L_2$ *distance.*

*Proof.* The linear mapping $x \mapsto W_k x$ is $\|W_k\|_2$-Lipschitz. Combined with the Lipschitz of activation function we have $\sigma(W_k x)$ is also $\|W_k\|_2$-Lipschitz. Then the composition of different layers has Lipschitz constant $\Pi_k M_k$. The Lemma follows by adding the Lipschitz of the last linear mapping. $\square$

Thus, we have

$$L(\mathcal{F}_{so}) \leq M(\alpha)\Pi_{k=1}^{K_{so}} M(k).$$

By Theorem 7,

$$\max_{\mathbf{q}\in\mathcal{Q}} \hat{\mathfrak{G}}_{\mathbf{q}}(\mathcal{F}_{so}) = \tilde{\mathcal{O}}(\frac{D_Z\sqrt{K_{so}}M(\alpha)\Pi_{k=1}^{K_{so}} M(k)}{\sqrt{n_{so}}}).$$

Plug the above two equations into (4), we have

$$\frac{\hat{\mathfrak{G}}_{Tn_{so}}\left(\mathcal{F}_{so}^{\otimes T} \circ \mathcal{H}\right)}{\sqrt{Tn_{so}}} = \tilde{\mathcal{O}}\left(\frac{D_X}{(Tn_{so})^2} + M(\alpha)\Pi_{k=1}^{K_{so}} M(k)(\frac{\hat{\mathfrak{G}}_{Tn_{so}}(\mathcal{H})}{\sqrt{Tn_{so}}} + \frac{D_Z\sqrt{K_{so}}}{\sqrt{n_{so}}})\right),$$

where $D_X = \sup_{(h,f,x),(h',f',x')\in\mathcal{H}\times\mathcal{F}_{so}\times\mathcal{X}} \|h \circ f(x) - h' \circ f'(x')\|$.

**Lemma 2.** *The boundedness parameter* $D_X$ *satisfies* $D_X \leq D_Z M(\alpha)\Pi_{k=1}^{K_{so}} M(k)$.

*Proof.* The proof is given by induction. Let $r_k$ denote the vector-valued output of the $k$-th layer of the prediction function. First note that

$$D_X \leq 2\sup_{f\in\mathcal{F}_{so}, z\in\mathcal{Z}} \|f(z)\|^2 \leq 2M(\alpha)\|r_{K_{so}}\|^2.$$

For each output of the $k$-th layer, we have

$$\|r_k\|^2 = \|\sigma(W_k r_{k-1})\|^2 \leq \|W_k r_{k-1}\|_2^2 \leq \|W_k\|_2^2\|r_{k-1}\|_2^2,$$

where the first inequality is by the 1-Lipschitz of the activation function. By induction, we have

$$D_X \leq 2D_Z M(\alpha)\Pi_{k=1}^{K_{so}} M(k).$$

$\square$

Recall that $\mathcal{F}_{ta} = \mathcal{L} \circ \mathcal{M}_{K_{ta}-K_{so}-1} \circ (\mathcal{F}_{so}^{\otimes p})$ and the Lipschitz constant $L' \leq M(\alpha)\Pi_{K_{so}+2}^{K_{ta}} M(k)$. Using Theorem 3 and apply Lemma 2, we have

$$\mathcal{E}_{ta}\left(\hat{f}_{ta}, \hat{h}\right) =$$

$$\tilde{\mathcal{O}}\left(p\nu\Pi_{k=K_{so}+2}^{K_{ta}} M(k)\left(M(\alpha)\Pi_{k=1}^{K_{so}} M(k)(\frac{\hat{\mathfrak{G}}_{Tn_{so}}(\mathcal{H})}{\sqrt{Tn_{so}}} + \frac{D_Z\sqrt{K_{so}}}{\sqrt{n_{so}}})\right) + \frac{D_Z\sqrt{K_{ta}} \cdot M(\alpha)\Pi_{k=1}^{K_{ta}} M(k)}{\sqrt{n_{ta}}}\right).$$

# E  Lower bound results for the diversity of depth-1 NN

We first give the proof using ReLu activation function (Theorem 4), as the result is more intuitive before we extend the similar results to other activation functions.

*Proof.* As we consider arbitrary representation function and covariate distribution, for simplicity we write $X' = h^*(X)$ and $Y' = h(X)$.

We consider a subset of depth-1 neural networks with ReLu activation function: $\mathcal{F} = \{x \mapsto [\langle x, w \rangle - (1 - \epsilon/4)]_+ : \|x\|_2 \leq 1, \|w\| \leq 1\}$. Let $f_w$ be the function with parameter $w$. Consider $U \subset \{x : \|x\|_2 = 1\}$ such that $\langle u, v \rangle \leq 1 - \epsilon$ for all $u, v \in U, u \neq v$.

**Lemma 3.** *For any $\mathcal{T} \subset \mathcal{F}$, $|\mathcal{T}| \leq \lfloor |U|/2 \rfloor$, there exists a $V \subset U$, $|V| \geq \lfloor |U|/2 \rfloor$ such that any $f \in \mathcal{T}$, $f(v) = 0$ for all $v \in V$.*

*Proof.* For any set $\mathcal{T}$, let $U_\mathcal{T} = \{u : \exists t \in \mathcal{T}, u \in \arg\max_{u \in U} \langle u, f_t \rangle\}$ be a subset of $U$. Thus, $|U_\mathcal{T}| \leq T \leq \lfloor |U|/2 \rfloor$. Let $V = U \setminus U_\mathcal{T}$.

For any $f \in \mathcal{T}$, let $u_f$ be its closed point in $U$. Let $v_f$ be its closed point in $V$. Let $\theta_f$ be the angle between $u_f$ and $v_f$. By the definition of $U$, we have $cos(\theta_f) = \langle u_f, v_f \rangle \leq 1 - \epsilon$. We will show that $\langle f, v_f \rangle \leq 1 - \epsilon/4$.

Note that since $\langle f, v_f \rangle \leq \langle f, u_f \rangle$, we have the angle between $f$ and $v$ is larger than $\theta_f/2$. By the simple fact that $cos(\theta_f/2) \leq 1 - (1 - cos(\theta_f))/4$, we have $\langle f, v_f \rangle \leq 1 - \epsilon/4$. Thus, $f(v_f) = 0$ and $f(v) = 0$ for all $v \in V$. $\square$

For any set of prediction functions in source tasks, let $V$ be the set defined in the above lemma. Consider any $u \in U \setminus V$ and let $V' = U \setminus (V \cup u)$. By this construction, we have $f_u(u) = \epsilon/4$, while all $f \in \mathcal{T}, f(u) = 0$. Note that

$$\inf_{f \in \mathcal{F}} \frac{1}{|V'|} \sum_{x \in V'} (f_u(u) - f(x))^2 \geq \frac{|V'| - 1}{16|V'|} \epsilon^2 \geq \frac{1}{32} \epsilon^2,$$

while

$$\sum_{f \in \mathcal{T}} \frac{1}{|V'|} \sum_{x \in V'} (f(u) - f(x))^2 = 0.$$

Thus, we let $X' = u$ almost surely and $Y'$ follows a uniform distribution over $V'$. This is true when the covariate distribution is the same as $Y'$ and $h = x \mapsto x$ and $h^* = x \mapsto u$. Recalling the definition of diversity, we have

$$\inf_{f \in \mathcal{F}} \mathbb{E}_{X', Y'} (f_u(X') - f(Y'))^2 = \epsilon^2/32 \text{ and } \frac{1}{T} \sum_{f_t \in \mathcal{T}} \inf_{f'_s \in \mathcal{F}} \mathbb{E}_{X', Y'} (f_s(X') - f'_s(Y'))^2 = 0.$$

Note that the same result holds when the bias $b \leq -(1 - \epsilon/4)$. For general bounded $\|b\|_2 \leq 1$, one can add an extra coordinate in $x$ as an offset. $\square$

In Theorem 4, we show that in depth-1 neural network with ReLu activation function, we will need exponentially many source tasks to achieve diversity. Similar results can be shown for other non-linear activation functions that satisfies the following condition:

**Assumption 3.** *Let $\sigma : \mathbb{R} \mapsto \mathbb{R}$ be an activation function. We assume there exists $x_1, x_2 \in \mathbb{R}$, $x_1 > x_2$, such that $|\sigma(x_1)| \geq \sup_{x \leq x_2} |\sigma(x)| M$ for some $M > 0$.*

ReLu satisfies the assumption with any $M > 0$ for any $x_1 > 0$ and $x_2 \leq 0$. Also note that any continuous activation function that is lower bounded and increasing satisfies this assumption.

**Theorem 9.** *Let $\sigma$ satisfies the above assumption with $M$ for some $x_1$ and $x_2$. Let $\mathcal{F} = \{x \mapsto \sigma(8(x_1 - x_2)\langle x, w \rangle - 7x_1 + 8x_2)) : \|x\|_2 \leq 1, \|w\|_2 \leq 1\}$. Let $\mathcal{T} = \{f_1, \ldots, f_T\}$ be any set of depth-1 neural networks with ReLu activation in $\mathcal{F}$. If $T \leq 2^{d \log(2) - 1}$, there exists some representation $h^*, h' \in \mathcal{H}$, some distribution $P_X$ and a target function $f^*_{ta} \in \mathcal{F}$, such that*

$$\frac{\inf_{f \in \mathcal{F}} \mathcal{E}_{ta}(f, h')}{\inf_{\boldsymbol{f} \in \mathcal{F}} \mathcal{E}_{so}(\boldsymbol{f}, h')} \geq \frac{(M - 1)^2}{8}.$$

*Proof.* We follow the construction in the proof of Theorem 9, fix an $\epsilon = 1/2$ and let $U \subset \{x : \|x\|_2 = 1\}$ such that $\langle u, v \rangle \leq 1 - \epsilon$ for all $u, v \in U, u \neq v$.

For any source tasks set $\mathcal{T}$, let $U_\mathcal{T} = \{u : \exists t \in \mathcal{T}, u \in \arg\max_{u \in U} \langle u, f_t \rangle\}$ be a subset of $U$. Thus, $|U_\mathcal{T}| \leq T \leq \lfloor |U|/2 \rfloor$. Let $V = U \setminus U_\mathcal{T}$. For any $f \in \mathcal{T}, v \in V$, similarly to the previous argument, we have $\langle f, v \rangle \leq 1 - \epsilon/4 = 1/8$. Therefore, $\langle f, v \rangle \leq 1 - \epsilon/4 = 1/8 \leq x_2$.

For any set of prediction functions in source tasks, let $V$ be the set defined in the above lemma. Consider any $u \in U \setminus V$ and let $V' = U \setminus (V \cup u)$. By this construction, we have $f_u(u) = \sigma(x_1)$, while all $f \in \mathcal{T}, f(u) = \sigma(x_2)$. Note that

$$\inf_{f \in \mathcal{F}} \frac{1}{|V'|} \sum_{x \in V'} (f_u(u) - f(x))^2 \geq \frac{|V'| - 1}{|V'|} \geq \frac{1}{2}(\sigma(x_1) - \sigma(x_2))^2,$$

while

$$\sum_{f \in \mathcal{T}} \frac{1}{|V'|} \sum_{x \in V'} (f(u) - f(x))^2 \leq 4 \sup_{x \leq x_2} \sigma(x_2)^2.$$

Thus

$$\frac{\inf_{f \in \mathcal{F}} \frac{1}{|V'|} \sum_{x \in V'} (f_u(u) - f(x))^2}{\sum_{f \in \mathcal{T}} \frac{1}{|V'|} \sum_{x \in V'} (f(u) - f(x))^2} \geq \frac{(M-1)^2}{8}$$

$\square$

# F   Proof of Theorem 5

*Proof.* Since $\dim^s(\mathcal{F}^*)$ is at least $d_E$, for any set $\{f_1, \ldots, f_t\}$, there exists a $f_{t+1}$ that is $(\mathcal{F}^*, \epsilon)$-independent of $\{f_1, \ldots, f_t\}$. By definition, we have

$$\exists x_1, x_2 \in \mathcal{X}, \sum_{i=1}^{t} \|f_i(x_1) - f_i(x_2)\|_2^2 \leq \epsilon^2, \text{ while } \|f_{t+1}(x_1) - f_{t+1}(x_2)\|_2^2 \geq \epsilon^2.$$

We only need to construct appropriate data distribution $P_X$ and representation $g, g^*$ to finish the proof. As we do not make any assumption on $g, g^*$ and $P_X$, it would be simple to let $X_1 = g(X)$ and $X_2 = g^*(X)$.

We let the distribution of $X_1$ be the point mass on $x_1$. Let $X_2$ be the uniform distribution over $\{x_1, x_2\}$.

For the excess error of source tasks, we have

$$\inf_{f_1', \ldots, f_t'} \sum_{i=1}^{t} \mathbb{E}_{X_1, X_2} \|f_i'(X_1) - f_i(X_2)\|_2^2$$

$$\leq \sum_{i=1}^{t} \mathbb{E}_{X_1, X_2} \|f_i(X_1) - f_i(X_2)\|_2^2$$

$$= \sum_{i=1}^{t} \frac{1}{2} \|f_i(x_1) - f_i(x_2)\|_2^2 \leq \frac{\epsilon^2}{2}.$$

For the excess error of the target task $f_{t+1}$, we have

$$\inf_{f_{t+1}' \in \mathcal{F}} \mathbb{E}_X \|f_{t+1}'(X_1) - f_{t+1}(X_2)\|_2^2$$

$$= \inf_{f_{t+1}' \in \mathcal{F}} [\frac{1}{2} \|f_{t+1}'(x_1) - f_{t+1}(x_2)\|_2^2 + \frac{1}{2} \|f_{t+1}'(x_1) - f_{t+1}(x_1)\|_2^2]$$

$$\geq \inf_{a \in \mathbb{R}} [\frac{1}{2} \|a - f_{t+1}(x_2)\|_2^2 + \frac{1}{2} \|a - f_{t+1}(x_1)\|_2^2]$$

$$= \frac{1}{4} \|f_{t+1}(x_1) - f_{t+1}(x_2)\|_2^2 \geq \frac{\epsilon^2}{4}.$$

The statement follows. $\square$

# G  Extending to general loss functions

In all the above analyses, we assume the square loss function for both source and target tasks. We first show that diversity under square loss implies diversity under any convex loss function. Let $\nabla l(x, y)$ be the gradient of function $\nabla l(\cdot, y)$ evaluated at $x$.

**Lemma 4.** *Any task set $\mathcal{F}$ that is $(\nu, \mu)$-diverse over any prediction space under square loss is also $(\nu/c_1, \mu/c_1)$-diverse over the same space under loss $l$, if $l$ is $c_1$ strongly-convex and for all $x \in \mathcal{X}$*

$$\mathbb{E}[\nabla l(g^*(X), Y) \mid X = x] = 0 \tag{5}$$

*Proof.* Using the definition of the strongly convex and (5),

$$\mathbb{E}_{X,Y}[l(f_t \circ h(X), Y) - l(f_t^* \circ h^*(X), Y)]$$
$$\geq \mathbb{E}_{X,Y}[\nabla l(f_t^* \circ h^*(X), Y)^T (f_t^* \circ h^*(X) - f_t \circ h(X)) + c_1 \|f_t^* \circ h^*(X) - f_t \circ h(X)\|_2^2]$$
$$= c_1 \mathbb{E}_{X,Y}[\|f_t^* \circ h^*(X) - f_t \circ h(X)\|_2^2],$$

which is the generalization error under the square loss. $\qquad\square$

Note that Equation (5), is a common assumption made in various analyses of stochastic gradient descent (Jin et al., 2021).

On the other direction, we show that any established diversity over the target task with square loss also implies the diversity over the same target task with any loss $l$ if $\nabla^2 l \succ c_2 I$ for some $c_2 > 0$.

**Lemma 5.** *Any task set $\mathcal{F}$ that is $(\nu, \mu)$-diverse over a target prediction space under square loss is also $(\nu c_2, \mu c_2)$-diverse over the same space under loss $l$, if $\nabla^2 l(\cdot, y) \succ c_2 I$ for all $y \in \mathcal{Y}_{ta}$ and for all $x \in \mathcal{X}$ we have $\mathbb{E}[\nabla l(g^*(X), Y) \mid X = x] = 0$.*

*Proof.* The proof is the same as the proof above except for changing the direction of inequality. Using the definition of the strongly convex and (5),

$$\mathbb{E}_{X,Y}[l(f_t \circ h(X), Y) - l(f_t^* \circ h^*(X), Y)]$$
$$\leq \mathbb{E}_{X,Y}[\nabla l(f_t^* \circ h^*(X), Y)^T (f_t^* \circ h^*(X) - f_t \circ h(X)) + c_2 \|f_t^* \circ h^*(X) - f_t \circ h(X)\|_2^2]$$
$$= c_2 \mathbb{E}_{X,Y}[\|f_t^* \circ h^*(X) - f_t \circ h(X)\|_2^2],$$

which is the generalization error under the square loss. $\qquad\square$

# H  Missing proofs in Section 6

Assume we have $T$ tasks, which is $(\nu, \mu)$-diverse over $\mathcal{F}_{so}$, and $\mathcal{Y}_{so} \subset \mathbb{R}$. Then we can construct a new source task $so$ with multivariate outputs, i.e. $\mathcal{Y}_{so} \subset \mathbb{R}^T$, such that $\mathcal{H}_{so} = \mathcal{F}_{so}^{\otimes T}$ and each dimension $k$ on the output, given an input $x$, is generated by

$$Y_k(X) = f_k^* \circ h^*(X) + \epsilon.$$

Intuitively, this task is equivalent to $T$ source tasks of a single output, which is formally described in the following Theorem.

**Theorem 10.** *Let $so$ be a source task with $\mathcal{Y}_{so} \subset \mathbb{R}^K$ and $f_{so}^*(\cdot) = (m_1^*(\cdot), \ldots, m_K^*(\cdot))$ for some class $\mathcal{M} : \mathcal{Z} \mapsto \mathbb{R}$. Then if the task set $t_1, \ldots, t_K$ with prediction functions $m_1^*, \ldots, m_K^*$ from hypothesis class $\mathcal{M}$ is $(\nu, \mu)$-diverse over $\mathcal{M}$, then so is $(\frac{\nu}{K}, \frac{\mu}{K})$-diverse over the same class.*

*Proof.* This can be derived directly from the definition of diversity. We use $t$ to denote the new task. By definition,

$$\inf_{f_{so} \in \mathcal{F}_{so}} \mathcal{E}_{so}(f_{so}, h) = \inf_{\boldsymbol{h}_{so}} \mathbb{E}_X \|(m_1 \circ (X), \ldots, m_K \circ h(X)) - (m_1^* \circ h^*(X), \ldots, m_K^* \circ h^*(X))\|_2^2$$

$$= \sum_{k=1}^{K} \inf_{m_k \in \mathcal{M}} \|m_k \circ h(X) - m_k^* \circ h^*(X)\|_2^2$$

$$= \sum_{k=1}^{K} \inf_{m_k \in \mathcal{M}} \mathcal{E}_{t_k}(f_{t_k}, h)$$

As $(t_1, \ldots, t_K)$ is $(\nu, \mu)$-diverse, we have

$$\frac{\sup_{m^* \in \mathcal{M}} \inf_{m \in \mathcal{M}} \mathcal{E}_{m^*}(m, h)}{\inf_{h_{so} \in \mathcal{F}_{so}} \mathcal{E}_{so}(f_{so}, h) + \mu/\nu} = \frac{1}{K} \frac{\sup_{m^* \in \mathcal{M}} \inf_{m \in \mathcal{M}} \mathcal{E}_{m^*}(m, h)}{\frac{1}{K} \sum_{k=1}^{K} \inf_{m_k \in \mathcal{M}} \mathcal{E}_{t_k}(h_k, h) + \frac{\mu}{\nu K}} \leq \frac{\nu}{K}.$$

$\square$

For the multiclass classification problem, we try to explain the success of the pretrained model on ImageNet, a single multi-class classification task. For a classification problem with $K$-levels, a common way is to train a model that outputs a $K$-dimensional vector, upon which a Softmax function is applied to give the final classification result. A popular choice of the loss function is the cross-entropy loss.

Now we formally introduce our model. Let the Softmax function be $q : \mathbb{R}^K \mapsto [0, 1]^K$. Assume our response variable $y \in \mathbb{R}^K$ is sampled from a multinomial distribution with mean function $q(f_{so}^* \circ h^*(x)) \in [0, 1]^K$, where $h^* \in \mathcal{H} : \mathcal{X} \mapsto \mathcal{Z}$ and $f_{so}^* \in \mathcal{F}_{so} : \mathcal{Z} \mapsto \mathbb{R}^K$. We use the cross-entropy loss $l : [0, 1]^K \times [0, 1]^K \mapsto \mathbb{R}, l(p, q) = -\sum_{k=1}^{K} p_k \log(q_k)$.

**Assumption 4.** *[Boundedness] We assume that any $f \circ h(x) \in \mathcal{F}_{so} \times \mathcal{H}$ is bounded in $[-\log(B), \log(B)]$ for some constant positive $B$. We also assume the true function $\min_k U(f^* \circ h^*(x))_k \geq 1/B_*$ for some $B_* > 0$.*

**Theorem 11.** *Under Assumption 4, a $K$-class classification problem with $f_{so}^*(\cdot) = (m_1^*(\cdot), \ldots, m_K^*(\cdot))$ for some $m_1^*, \ldots, m_K^* \in \mathcal{M}$ and Softmax-cross-entropy loss function is $(2B_*^2 B^4 \nu, B_*^2 \mu)$-diverse over any the function class $\mathcal{M}$ as long as $f_{so}^*$ with $L_2$ loss is $(\nu, \mu)$-diverse over $\mathcal{M}$.*

*Proof.* We consider any target task with prediction function from $\mathcal{M}^{\otimes K'}$. Let $U : \mathbb{R}^{K'} \mapsto [0, 1]^{K'}$ be the softmax function. We first try to remove the cross-entropy loss. By definition, the generalization error of any $f \circ h \in \mathcal{M}^{\otimes K'} \times \mathcal{H}$ is

$$\mathcal{E}_{ta}(f \circ h) - \mathcal{E}_{ta}(h^* \circ h^*)$$

$$= \mathbb{E}_{X,Y}[-\sum_{i=1}^{K'} \mathbb{1}(Y = i) \log(\frac{U(f \circ h)}{U(f^* \circ h^*)})]$$

$$= \mathbb{E}_X[-\sum_{i=1}^{K'} U(f^* \circ h^*) \log(\frac{U(f \circ h)}{U(f^* \circ h^*)})], \tag{6}$$

which gives us the KL-divergence between two distributions $U(f \circ h)$ and $U(f^* \circ h^*)$.

**Lemma 6.** *For any two discrete distributions $p, q \in [0, 1]^K$, we have*

$$KL(p, q) \geq \frac{1}{2}(\sum_{i=1}^{K} |p - q|)^2 \geq \frac{1}{2} \sum_{i=1}^{K} (p_i - q_i)^2.$$

*On the other hand, if $\min_i p_i \geq b$ for some positive $b$, then*

$$KL(p, q) \leq \frac{1}{b^2} \sum_{i=1}^{K'} (p_i - q_i)^2.$$

*Proof.* The first inequality is from Theorem 2 in Dragomir and Gluscevic (2000). The second inequality is by simple calculus. $\square$

*By the assumption 4, we have that for any $h, g, x$,*

$$U(f \circ h(x))_i \in [\frac{1}{KB^2}, \frac{1}{1 + (K-1)/B^2}].$$

*We also have $\sum_{i=1}^{K} \exp(f \circ h(x)_i) \in [K/B, KB]$. To proceed,*

$$\frac{\sup_{f_{ta}^* \in \mathcal{M}^{\otimes K_{ta}}} \inf_{\hat{f}_{ta}} \mathcal{E}_{ta}(\hat{f}_{ta} \circ h) - \mathcal{E}_{ta}(f_{ta}^* \circ h^*)}{\inf_{\hat{f}_{so} \in \mathcal{M}^{\otimes K_{so}}} \mathcal{E}_{so}(\hat{f}_{so} \circ h) - \mathcal{E}_{so}(f_{so}^* \circ h^*) + \frac{B_*^2 \mu}{2B_*^2 B^4 \nu}}$$

*(Applying (6) and Lemma 6)*

$$\leq 2B_*^2 \frac{\sup_{f_{ta}^* \in \mathcal{M}^{\otimes K_{ta}}} \inf_{\hat{f}_{ta}} \mathbb{E}_X \|U(\hat{f}_{ta} \circ h(X)) - U(f_{ta}^* \circ h^*(X))\|_2^2}{\inf_{\hat{f}_{so} \in \mathcal{M}^{\otimes K_{so}}} \mathbb{E}_X \|U(\hat{f}_{so} \circ h(X)) - U(f_{so}^* \circ h^*(X))\|_2^2 + \frac{\mu}{B^4 \nu}}$$

*(Using the boundedness of $\displaystyle\sum_{i=1}^{K} \exp(f \circ h(x)_i)$)*

$$\leq 2B_*^2 B^2 \frac{\sup_{f_{ta}^* \in \mathcal{M}^{\otimes K_{ta}}} \inf_{\hat{f}_{ta}} \mathbb{E}_X \|\exp(\hat{f}_{ta} \circ h(X)) - \exp(f_{ta}^* \circ h^*(X))\|_2^2}{\inf_{\hat{f}_{so} \in \mathcal{M}^{\otimes K_{so}}} \mathbb{E}_X \|\exp(\hat{f}_{so} \circ h(X)) - \exp(f_{so}^* \circ h^*(X))\|_2^2 + \frac{\mu}{B^2 \nu}}$$

*(Using the Lipschitz and convexity of $exp$)*

$$\leq 2B_*^2 B^4 \frac{\sup_{f_{ta}^* \in \mathcal{M}^{\otimes K_{ta}}} \inf_{\hat{f}_{ta}} \mathbb{E}_X \|\hat{f}_{ta} \circ h(X) - f_{ta}^* \circ h^*(X)\|_2^2}{\inf_{\hat{f}_{so} \in \mathcal{M}^{\otimes K_{so}}} \mathbb{E}_X \|\hat{f}_{so} \circ h(X) - f_{so}^* \circ h^*(X)\|_2^2 + \mu/\nu}$$

$$\leq 2B_*^2 B^4 \nu.$$

*The diversity follows.*

$\square$

# I   Experimental details

Each dimension of inputs is generated from $\mathcal{N}(0, 1)$. We use Adam with default parameters for all the training with a learning rate 0.001. We choose ReLu as the activation function.

**True parameters.**   The true parameters are initialized in the following way. All the biases are set by 0. The weights in the shared representation are sampled from $\mathcal{N}(0, 1/\sqrt{n_u})$. The weights in the prediction function for the source task are set to be orthonormal when $K_{so} = 1$ and $p \leq n_u$. For the target prediction function or source prediction function if $K_{so} > 1$, the weights are sampled from $\mathcal{N}(0, 1/\sqrt{n_u})$ as in the representation part.

**Hyperparameters.**   Without further mentioning, we use the number of hidden units, $n_u = 4$, input dimension $p = 4$, $K = 5$, $K_{ta} = K_{so} = 1$, the number of observations $n_{so} = 1000$ and $n_{ta} = 100$ by default. Note that since $p$ is set to be 4 by default, equivalently we will have $n_{so} \cdot p = 4000$ observations.