# OpenReview forum: "Representation Learning Beyond Linear Prediction Functions"
_NeurIPS.cc/2021/Conference — NeurIPS 2021 Spotlight_

### Official Review · Reviewer_5xzU · 2021-07-10

**Rating:** 7
**Confidence:** 3

**Summary:**

This paper studies the importance of diversity in representation learning. The novelty over previous work is that this work studies the case where the prediction classes for the source and target can be different and nonlinear.

This paper first shows that diversity over the source prediction class $F_{so}$ implies diversity over the target prediction class $F_{ta}$, both of which could be nonlinear, as long as $F_{ta}$ is more complex than $F_{so}$. On the other hand, if $F_{so} \not\subset F_{ta}$, then 0-bias diversity cannot hold. When diversity does hold, error bounds on the target task are derived in terms of the diversity constants.

The second contribution of the paper is a negative result for nonlinear $F_{so}$: the number of tasks required to achieve $(\nu, 0)$-diversity is lower bounded by $\Omega(2^d)$ for 1-layer NN, or by the eluder dimension for the general case which is known to grow exponentially.

Finally, the theoretical results are verifies empirically on synthetic data. Different than the multi-task setup for theory, the experiments are on a single task with multiple outputs. The results show that diversity can still be achieved in this setting, with the representation learning performance improving with the diversity as desired.

**Limitations And Societal Impact:**

The authors discussed potential future direction of taking into consideration the optimization algorithm, and potential social impact of causing unfairness if the tasks were unfairly chosen for under-represented groups.

**Main Review:**

**Significance and novelty of the contribution**: This paper generalizes the analysis of diversity to the case when the source and target prediction classes are nonlinear and nonidentical. This is an important problem to study, though the technical novelty is somewhat limited.

**Question**:
* The exponential lower bounds implied by Theorem 4 & 5 seem to be based on the assumption of $\mu = 0$. This seems to fit the  diversity definition on page 1, not the $(\nu, \mu)$-diversity in Def 1 where $\mu$ can be non-zero, which would mean the exponential lower bound no longer applies.
* For empirically verify the effect of diversity (Fig 1(c)), the value of $p$ seems too small to extract a confidence conclusion; moreover, the transfer performance is worse than the baseline in 3 out of 4 choices of $p$. Could you provide results with larger scale dataset (e.g. higher values of $p$ and more data points) please?

**Clarify of writing**: the paper is reasonably structured overall, though some parts would benefit from some rephrasing.
  - Typo: bottom of page 1, the diversity condition: $f_{ta}^*$ is not in the definition.
  - The paragraph after line 26: "at least $d$ source tasks": please add references.
  - Appendix B doesn't read well; e.g. line 400 (appendix) says "Now using Assumption 1", but the equation that follows doesn't use it.
  - Typo: the assumption on weight norms right above line 165.
  - Line 206: please provide references.
  - Thm 5: please provide a definition of the dual class.
  - Line 214 says that Thm 5 is consistent with Theorem 9, which is not mentioned in the main text. Perhaps change this to "consistent with Theorem 4".
  - Line 220: this should be $(T, \mu)$-diverse; $\nu$ is defined differently in Tripuraneni et. al. than in this work. Moreover, the result in Tripuraneni et. al. has some quantifiers on $\mathcal{F}$; please make this clear.
  - Typo: Proposition 2: "$w(f_1), ..., w(f_{d_i})$ span $\mathbb{R}^{d_i}$".

======
Post-rebuttal update: I thank the authors for their responses and have raised my score.

**Time Spent Reviewing:**

4

---

> ### Author Response · Authors · 2021-08-07
> **Response to two questions**
>
> We thank the reviewer for the fruitful comments. We accept all the writing issues and will fix it in the camera-ready version. Here are our responses to the two questions.
>
> $\textit{First question.}$ Indeed, Theorem 4 and 5 are discussing the case when $\mu = 0$. However, we argue that it is not interesting to discuss large $\mu$ at all. For example, in the context of Theorem 4, it requires that $\mu > \epsilon^2/32$ to satisfy the definition of diversity (Definition 1). By Theorem 1, the generalization error of target task will have a loose bound of $\epsilon^2/32$, which is much worse than the source tasks loss $0$, which means we can not have a good bound on diversity parameters to upper bound target. That is to say, sometimes even if the diversity holds (possibly with a large $\mu$), the generalization error bound can still be loose.
>
> Moreover, the results in Theorem 4 and 5 directly target on the generalization error without the need to use the definition of diversity. In Theorem 4, there exists a case such that
> $\inf_{f \in \mathcal{F}} \mathcal{E}_{t a}(f, h^{\prime})=\epsilon^{2} / 32$
>
> while $\inf_{\mathbf{f} \in \mathcal{F} \otimes T} \mathcal{E}_{s o}\left(\mathbf{f}, h^{\prime}\right)=0$, which means it is possible to have an irreducible $\epsilon^2/32$ loss on a target task even if one minimizes the expected loss on source tasks to 0.
>
> $\textit{Second question.}$ Yes, we can provide higher values of $p$. It is possible that the transfer performance is worse than baseline since when only few source tasks are available, the linear coefficients of the target task could be almost orthogonal to the coefficients of source tasks. Thus the representation learnt could be useless.

---

> > ### Author Response · Authors · 2021-08-08
> > **Figure 1 (c) with a larger p = 8**
> >
> > We reproduced the experiments in Figure 1 (c) with p = 8. We also set the number of hidden units to be 8. The numbers are larger than these in the original figure since the size of the problem changes. Here are the average MSE's for number of outputs $ = 1, \dots, 8$ and for baseline:
> > 0.6103, 0.5715, 0.4972, 0.5009, 0.4964, 0.5053, 0.5058, 0.4831, 0.4962
> >
> > The standard deviations are
> > 1.2015, 1.0716, 0.9564, 0.9511, 0.9547, 0.9658, 0.9651, 0.9427, 0.9567
> >
> > The overall trend is the same. We have worse performance when $p$ is small and the performances are roughly the same for large $p$.

---

> > > ### Comment · Reviewer_5xzU · 2021-08-22
> > > **Thank you for your response**
> > >
> > > I thank the authors for their clarification and the additional results. I will raise my score to 7.

---

### Official Review · Reviewer_RHzy · 2021-07-16

**Rating:** 7
**Confidence:** 4

**Summary:**

The paper studies theory of representation learning. It differs from existing papers by considering scenarios when (a) the source and the target function class are different, and (b) non-linear prediction functions are used. There are three main results. The first result provides an upper bound for the generalization error of the target task if the source tasks are diverse. The second result shows the hardness of being diverse if the source tasks use nonlinear prediction functions, by relating diversity and eluder dimension. And the third result connects diversity and generalized rank.

**Limitations And Societal Impact:**

See the section above.

**Main Review:**

Significance: The paper studies two meaningful extensions of representation learning theory as outlined in the summary. Both extensions are inspired by common practice in deep representation learning. And therefore studying these extensions is a necessary and important step to breach the gap between theory and practice.

Originality: Across all the results presented in the paper, the most novel and exciting result is the link between diversity and eluder dimension (the second result on hardness). As noted by the author, this paper is the first to discover such a connection. The remaining results in the paper are built upon existing works, with little novelty in both the results and the proof techniques. The lack of a literature review makes it hard for readers to distinguish the contributions of the paper and various known results in the field.

Technical correctness:
-	To the best of my knowledge, I have checked the proof and finds no glaring issues with the technical theorems and lemmas in the paper. Therefore, I would like to focus the discussion of this section to understand more about the proof techniques and the implications of the theoretical results for practitioners.
-	Proof technique: Firstly, from what I understand, Theorem 1 is almost a direct result of the diversity assumption (Definition 1). As noted by the authors, the improvement of Theorem 1 as compared to standard generalization bound in supervised learning (e.g., in [2]) is in the third term. However, this improvement is achieved by fixing a representation \hat{h} and apply Theorem 6 (built upon [2]) over the predictor function space. Secondly, the hardness result for the general function class in Theorem 5 is, again, almost a direct result of the eluder dimension definition when t is less than the eluder dimension. Thirdly, the upper bound with the generalized rank result is a direct application of an existing Lemma in [1] (as noted by the author). To this end, I think that the three results are not so significant in terms of proof technique.
-	Implication for practitioners: Although these settings studied in the paper are inspired by common practice, the paper does not discuss (and I find it hard to interpret) the implications of these results. For example, many of the results in the paper rely on the assumption that the source tasks are diverse. And Proposition 2 is the only result that tells us how to achieve diversity through the notion of generalized rank. However, as noted by the authors, even upper bounding the generalized rank is hard. In the end, the paper does not discuss a concrete method to achieve diversity in the source task. For this reason, I find it challenging to use the theory presented in this paper as guidance for empirical practice.

Experiments: In terms of empirical studies, I appreciate the various experiments exploring the behavior of representation learning. One major gap between the developed theory and practice is that empirically, we cannot find the minimizers, but instead arbitrary local minimizers via gradient-based methods. Therefore, I understand that mismatch between theory and practice is unavoidable. I find the introduction of a single source task with multiple outputs and its connection to multiple source tasks a bit arbitrary and unnecessary. Other than that, I am satisfied with the empirical results.

Clarity: The main paper is clear and well-written. I can follow and understand the ideas most of the time. Notations are consistent and correct. However, there is no section dedicated to discussing related works. The lack of such discussion makes it very hard to distinguish the contributions of the papers and known results. I understand that it could be due to page limits that the authors decide to remove the related work section and just refer to existing work throughout the development of the paper. However, I would appreciate it if the paper discusses existing work, as well as connects the presented results to existing results in the literature in the supplementary material.

Minor comments:
-	While the paper mentions that Theorem 6 is achieved by applying the standard bounded difference inequality from [2], and provides no proof. To me, even after glancing through [2], I could not quite understand how to get Theorem 6. I would appreciate it if the paper elaborate more or include a short proof of Theorem 6.
-	In Theorem 4, should the left infimum be taken over \mathcal{F}^{\otimes T}?
-	In Theorem 4, where does h^\ast show up?
-	In Theorem 5, the notion of dual-class of function space is not defined.
-	In definition 3, should the \mathcal{R} be just R? Also, the identity rank notation is not consistent: id-rk and id-rank.
-	On line 287, should the K_{so}=1 only and not 2, 3, since K_{so}=2, 3 give negative results.

[1] On the Theory of Transfer Learning: The Importance of Task Diversity.

[2] Rademacher and Gaussian Complexities: Risk Bounds and Structural Results.

======== After rebuttal ========

Thank you for the response. I am willing to increase my score from 6 to 7. As discussed, please include a related work section in the future version of the paper.

**Time Spent Reviewing:**

7

---

> ### Author Response · Authors · 2021-08-07
> **Response to Reviewer 3**
>
> We appreciate all the insightful comments.
>
> $\textit{Technical contributions.}$ We do follow the previous literature, while we consider a new and novel setting as the reviewer also acknowledged. There are some nontrivial proofs in the Supplementary when we generalize the result on ReLU to general activation functions and the results to general loss functions, especially cross-entropy for classification problem.
>
> $\textit{How to achieve diversity.}$ We agree that our paper lacks discussions on how to achieve diversity. Diversity is easy to achieve for linear prediction functions. One can simply choose source tasks such that the matrix formed by their linear coefficients is not degenerated. The most diverse set is to choose orthogonal coefficients. Beyond linear prediction functions, diversity is generally hard to achieve as indicated by Theorem 4. However, if the linear mappings are a subset of the function class, one can always choose the diverse linear mappings and apply Theorem 3. Empirically, this is a curriculum learning problem (scheduling tasks to achieve better learning benefits). Different empirical algorithms have been studied. For example, Alex et al. (2017) [1] and Ksenia et al. (2017) [2]  predict the gain in losses after one extra sample from each task and select the task that gives the highest local improvement. One can show that if the prediction is accurate, this strategy will lead to the most diverse source tasks for linear prediction functions.
>
> $\textit{Existing works.}$ Thanks for the comments. We did not add the related work section due to the page limit. We will discuss this in the camera-ready version.
>
> $\textit{Proofs of Theorem 6.}$ We apologize that Theorem 6 is under the assumption that the response is between $[0, 1]$ (one can always normalize the response to have it between [0, 1]). Here is a proof sketch of Theorem 6. Theorem 6 is stated in terms of Gaussian complexity. It is more common to use Radamecher complexity, which can be upper bounded by $\sqrt{2\pi}$ of the corresponding Gaussian complexity. For the generalization bound in terms of Rademacher complexity, Theorem 26.5 of Shai et al. (2014) has a full proof possibly easier to follow. Then we get rid of the loss function by the contraction lemma, which leads to Theorem 6.
>
> $\textit{In Theorem 4, where does $h^\ast$ show up?}$ Both $\mathcal{E}_{so}$ and $\mathcal{E}_{ta}$ depends on $h^\ast$. The notation is hidden for simplicity. See Section 2 for details.
>
> $\textit{Definition of dual space.}$ Our definition of dual space exchanges the models and inputs. For any $\mathcal{F}: \mathcal{X} \mapsto \mathcal{Y}$, its dual space is $\mathcal{F}^*:\mathcal{F} \mapsto \mathcal{Y}$ such that $\mathcal{F}^* = \\{g_x: g_x(f) = f(x), \forall x \in \mathcal{X}\\}$.
>
> [1] Graves, Alex, et al. "Automated curriculum learning for neural networks." international conference on machine learning. PMLR, 2017.
> [2] Konyushkova, Ksenia, Raphael Sznitman, and Pascal Fua. "Learning active learning from data." arXiv preprint arXiv:1703.03365 (2017).
> [3] Shalev-Shwartz, Shai, and Shai Ben-David. Understanding machine learning: From theory to algorithms. Cambridge university press, 2014.

---

### Official Review · Reviewer_ts9h · 2021-07-21

**Rating:** 7
**Confidence:** 2

**Summary:**

This paper studies a transferred representation learning problem. In the considered setting, there are a set of source tasks and a target task. Each task consists of representation learning and prediction. The assumption is that all tasks share the same best representation function. The core question is: if we run ERM over source tasks, is the learned representation a good representation for the target task, and what is the generalization error for the target task? The authors quantify the performance through the Gaussian complexity of the function classes, and a quantity called "diversity", which relates the generalization error between the source tasks and the target task. Then the authors discusses several special cases, including "source tasks are more complex than the target tasks", "source tasks are simply learning representation", "source tasks are representation followed by non-linear functions". The main message is the simpler the source tasks are, the better chance transfer representation learning will succeed.

**Limitations And Societal Impact:**

Yes.

**Main Review:**

This paper provides an interesting and clean viewpoint to quantify the performance of transfer representation learning.  The arguments in the paper look convincing and intuitive, while still providing new insight. The experimental results are also largely aligned with the theory. For the non-linear source task part, the lower bound using eluder dimension or the upper bound using generalized rank seem very pessimistic to me, because they are usually large beyond linear functions. The lower bound proof is also in a worst-case sense and may not reflect the reality. It would be good if the authors can provide discussions on this, and comment on whether those bounds can be improved using other complexity measures or considering more benign cases.

**Time Spent Reviewing:**

8

---

> ### Author Response · Authors · 2021-08-07
> **Benign case that may have better lower bound**
>
> We thanks the reviewer for the good question on how bad the worst-case could be. First we want to emphasize that our worst-case lower bound is to provide a general understanding that how hard it might be to achieve diversity in non-linear case. Some stronger assumptions can help us escape the exponentially large lower bound. As studied in Maurer et al. (2016) [1], if the new tasks are sampled from a fixed distribution, we can have a tighter upper bound. We believe the lower bound can also be improved in this setting and the right complexity measure is star number [2] (though no research has been done on this topic). Li, Gene, et al. also showed that star number is strictly smaller than eluder dimension and the gap can be exponentially large.
>
> [1] Maurer, Andreas, Massimiliano Pontil, and Bernardino Romera-Paredes. "The benefit of multitask representation learning." Journal of Machine Learning Research 17.81 (2016): 1-32.
> [2] Hanneke, Steve, and Liu Yang. "Minimax analysis of active learning." J. Mach. Learn. Res. 16.12 (2015): 3487-3602.

---

### Official Review · Reviewer_WX82 · 2021-07-28

**Rating:** 7
**Confidence:** 4

**Summary:**

The paper studies representation learning in a setting where the source tasks and the target task use different prediction function spaces beyond linear functions, and asks the following question: how many source tasks are required to achieve *diversity* (an important quantity related to the generalization ability from source tasks to the target task)?

The paper first shows that diversity over the source tasks' prediction function space implies diversity over the target task's prediction function space. As a result, even when the target task uses multi-layer neural networks as prediction functions, generalization is still guaranteed as long as the source tasks use linear prediction functions. However, when the source tasks use nonlinear prediction functions, there exist some fundamental limits for generalization: the paper proves some hardness results, including a novel lower bound on the number of required source tasks in terms of the eluder dimension. The paper further provides an upper bound in terms of the generalized rank. Furthermore, the paper gives some theoretical results indicating that simpler tasks generalize better. Such insights are validated by numerical experiments.

**Ethical Concerns:**

The paper has potential consequences on fairness, which have been clearly acknowledged in Section 7 of the paper.

**Limitations And Societal Impact:**

When the source tasks use nonlinear prediction functions, the paper provides an upper bound stated in terms of the generalized rank. Is it possible to provide an upper bound based on the eluder dimension or a lower bound based on the generalized rank? If not, then pointing out the main obstacles may be helpful. Also, understanding the roles, differences and interpretations of the two complexity measures (or some other potentially useful complexity measures) seem like an important future direction.

**Main Review:**

The paper is well-written. The considered problem is not only theoretically interesting but also practically relevant. I checked some key parts of the proofs, and believe that the claimed results are correct. The connection between representation learning and the eluder dimension is particular interesting --- to my knowledge, the eluder dimension is mainly studied in the bandit and RL literature, with much more upper bounds proved than lower bounds. The connection between representation learning and the eluder dimension discovered in this paper is novel and may lead to deeper understanding of related fields. I also like the fact that many theoretical results in this paper are complemented by numerical experiments (though not in exactly the same setups).

**Time Spent Reviewing:**

6

---

> ### Author Response · Authors · 2021-08-07
> **Upper bound with Eluder dimension or lower bound with generalized rank**
>
> We thank the reviewer for supporting our paper. Indeed, we agree that it is a very important direction to discuss how to close the gap. As pointed out by Li et al. (2021)[1], generalized rank can be exponentially larger than the Eluder dimension and they compared the difference between many related complexity measure. It might be possible to show an upper bound with Eluder dimension. The main obstacle is that the definition of Eluder dimension is states in terms of any two $f, \tilde{f} \in \mathcal{F}$, while to bound the generalization error, we need the it holds for an expectation over $\mathcal{F}$ (See definition 2). In our problem, since we consider the dual space, we actually need the same definition over any two distributions over $\mathcal{X}$.
>
> [1] Li, Gene, et al. "Eluder dimension and generalized rank." arXiv preprint arXiv:2104.06970 (2021).

---

### Decision · Program_Chairs · 2021-09-27

**Decision:**

Accept (Spotlight)

**Comment:**

The reviewers agreed that this paper studies a practically relevant problem (transfer/representation learning) and provides novel upper and lower bounds on transfer learning performance, as well as experimental results to complement the theoretical guarantees. While many of the technical tools (e.g., eluder dimension) will be familiar within research on contextual bandits and reinforcement learning, their use in this context is novel, and is likely to find broader use. The results here also suggest a number of interesting new follow-up directions. The paper is also well-written.